# Effects of clouds and aerosols on downwelling surface solar irradiance nowcasting and sort-term forecasting

Kyriakoula Papachristopoulou[1,2], Ilias Fountoulakis[3,2], Alkiviadis F. Bais[4], Basil E. Psiloglou[5], Nikolaos Papadimitriou[6], Ioannis-Panagiotis Raptis[1], Andreas Kazantzidis[6], Charalampos Kontoes[2], Maria Hatzaki[1], Stelios Kazadzis[7]

[1]Laboratory of Climatology and Atmospheric Environment, Section of Geography and Climatology, Department of Geology and Geoenvironment, National and Kapodistrian University of Athens, Athens, GR-15784, Greece
[2]Institute for Astronomy, Astrophysics, Space Applications and Remote Sensing, National Observatory of Athens (IAASARS/NOA), Athens, GR-15236, Greece
[3]Research Centre for Atmospheric Physics and Climatology, Academy of Athens, Athens, Greece
[4]Laboratory of Atmospheric Physics, Aristotle University of Thessaloniki, Thessaloniki, Greece
[5]Institute for Environmental Research and Sustainable Development, National Observatory of Athens (IERSD/NOA), Athens, GR-15236, Greece
[6]Laboratory of Atmospheric Physics, Department of Physics, University of Patras, GR 26500, Patras, Greece
[7]Physikalisch-Meteorologisches Observatorium Davos, World Radiation Center (PMOD/WRC), Davos 7260, Switzerland

*Correspondence to*: Kyriakoula Papachristopoulou (kpapachr@noa.gr)

**Abstract.** Solar irradiance nowcasting and short-term forecasting are important tools for the integration of solar plants into the electricity grid. Understanding the role of clouds and aerosols in those techniques is essential for improving their accuracy. In this study, we introduce the improvements in the existing nowcasting/short-term forecasting operational systems SENSE/NextSENSE, based on a new configuration and by the upgrade of cloud and aerosol inputs and, we also investigate the limitations of such model evaluation with surface-based sensors due to cloud effects. We assess the real-time estimates of surface global horizontal irradiance (GHI) produced by the improved SENSE2 operational system at high spatial and temporal resolution (~5 km, 15 min) for a domain including Europe and Middle East-North Africa (MENA) region and the short-term forecasts of GHI up to 3 hours ahead by the NextSENSE2 system, against ground-based measurements from 10 stations across the models' domain, for a whole year (2017).

Results for instantaneous (every 15 minutes) comparisons show that the GHI estimates are within +/-50 W/m$^2$ (or +/-10%) of the measured GHI for 61% of the cases, after the new model configuration and a proposed bias correction. The bias ranges between -12 W/m$^2$ to 23 W/m$^2$ (or -2% to 6.1%) with mean value 11.3 W/m$^2$ (2.3%). The correlation coefficient is between 0.83 to 0.96 with mean value 0.93. Statistics are significantly improved when integrating in daily and monthly scales (mean bias 3.3 W/m$^2$ and 2.7 W/m$^2$, respectively). We demonstrate that the main overestimation of the SENSE2 GHI is linked with the uncertainties of the cloud related information within the satellite pixel, while relatively low overestimation linked with aerosol optical depth (AOD) forecasts (derived from Copernicus Atmospheric Monitoring Service - CAMS) is reported for cloudless sky GHI. The highest deviations for instantaneous comparisons are associated with cloudy atmospheric conditions with clouds obscuring the sun over the ground-based station. Thus, they are much more linked with satellite/ground-based

comparison limitations than the actual model performance. The NextSENSE2 GHI forecasts based on the cloud motion vector (CMV) model, outperform the persistence forecasting method, which assumes the same cloud conditions for the future time steps. The forecasting skill (FS) of the CMV based model compared to the persistence approach increases with cloudiness (FS up to ~20%), linked mostly to periods with changes in cloudiness, that persistence, by definition, fails to predict. Our results can be useful for further studies on satellite-based solar model evaluations and, in general, for the operational implementation of solar energy nowcasting and short-term forecasting, supporting solar energy production and management.

## 1 Introduction

Climate change mitigation along with energy production in a sustainable manner could be addressed with the deployment of renewable energy technologies (Edenhofer et al., 2011; Pörtner et al., 2022). Diverse technologies of renewable energy are investigated worldwide, and their deployment has been increasing, with solar energy markets growing rapidly, with a prospect to be the major source of energy supply in next decades (Arvizu et al., 2011; IEA, 2022). Since solar energy resources are strongly affected by atmospheric conditions, they are highly variable in space and time. Therefore, there is a need for operational nowcasting and short-term solar forecasting for real time energy production, to better integrate solar energy exploitation technologies with national and international power systems. Under all-skies the availability of solar resources is primarily affected by clouds (e.g., Fountoulakis et al. 2021) and for clear-sky conditions it depends on the atmospheric composition with the most important variables being aerosols (e.g., Papachristopoulou et al., 2022) and water vapor (Yu et al., 2021). Among those variables, clouds and aerosols are characterized by large temporal and spatial variability which constitutes them as key variables for solar energy applications. The continuously improved earth observation (EO) data (satellite-based and atmospheric models) are exploited to produce in real time (nowcasting) accurate estimates of spectral surface solar radiation, with numerous applications apart from the solar energy sector (e.g., Qu et al., 2014; Thomas et al. 2016; Kosmopoulos et al., 2018) in different fields like human health (e.g., Kosmopoulos et al., 2021; Schenziger et al., 2023). To increase the accuracy of those nowcasting and forecasting tools, it is imperative to understand the spatiotemporal variability of cloud and aerosol properties in their implementation.

Solar resources assessment at a particular location is important for planning and management of solar energy technologies. The use of ground-based measurements of surface solar radiation is only available in a few locations with possible gaps in time. Those spatial and temporal gaps are filled by modelled estimates of surface solar radiation. Of particular importance are gridded surface solar radiation estimates with high spatial and temporal resolution and a large coverage (up to global scale), provided by satellite-based models or atmospheric models (e.g., an overview of those techniques in Sengupta et al., 2021). Geostationary satellite data due to their large area coverage and high temporal resolution are used to produce estimates of surface solar radiation both at real time as an operational service and to generate historical archives based on long term satellite measurements. Several methods exist for satellite estimates of surface solar irradiance. A well-established method considers cloud extinction through the cloud coverage index (Cano et al., 1986) or cloud index, calculated by normalized satellite

reflectances. Using the cloud index, the transmission factor or the clear sky index (also called cloud modification factor - CMF hereafter) is calculated, which finally multiplied with the results of a clear sky model to retrieve solar irradiance at the earth surface (Hammer et al., 2003). This is the general idea behind the HELIOSAT method (Cano et al., 1986; Hammer et al. 2003), widely used in various European research projects and applications. The derivative Heliosat-2 method (Rigollier et al., 2004) is launched in real time by the SoDa service to produce the HelioClim-3 database (Qu et al., 2014), a real time solar radiation database from February 2004 onwards. The more recent and most advanced version 5 (HC3v5) of the HelioClim-3 database (Thomas et al. 2016) combines the McClear clear-sky model (Lefevre et al., 2013; Gschwind et al., 2019) with cloud index values extracted from Meteosat Second Generation (MSG) satellite images. The Satellite Application Facility on Climate Monitoring (CM SAF) provide satellite-based estimates of surface solar radiation using data from Meteosat geostationary satellites. Currently, the third edition of the Surface Solar Radiation Data Set - Heliosat (SARAH-3, Pfeifroth et al. 2023a) covers the period 1983 - 2020 as climate data record (CDR) and is operationally extended as interim climate data record (ICDR) to the present with a delay of a few days. The retrieval algorithm MAGICSOL (Pfeifroth and Trentmann, 2023; Müller et al., 2015) is a combination of a modified Heliosat method to derive the effective cloud albedo (CAL) and the SPECMAGIC clear-sky model (Mueller et al., 2012). More available open access satellite-based surface solar radiation climatological datasets based on the cloud index method can be found in Müller and Pfeifroth (2022).

There are also fully physical models, that directly estimate surface solar radiation using radiative transfer models (RTM) and geophysical parameters as inputs for given atmospheric state including clouds (cloud and aerosol optical properties, total column values for water vapor and ozone content) and surface conditions. The combination of multi-channel information from geostationary satellites with cloud retrieval schemes provides cloud optical properties that can be explicitly used in RTM to account for cloud extinction and finally calculate surface solar radiation. Parameterizations or look-up-tables based on RTM simulations are used instead of direct radiative transfer calculations, to optimize the computational time for operational use of the models. This is the case for the Heliosat-4 method (Qu et al., 2017) which is used in Copernicus Radiation Service (CAMS Radiation Service) estimates of surface solar irradiance. Their Heliosat-4 method is composed of two models considering independently the clear sky and cloudy conditions. Specifically, the McClear model (Lefevre et al., 2013; Gschwind, et al., 2019) is used for calculations of cloud free irradiances and the McCloud model for calculating the extinction of irradiance by clouds (through the clear sky index), both based on look-up tables (LUTs) to speed up calculations. The input cloud properties of the current CAMS Radiation Service v4 are retrieved by the adapted APOLLO Next Generation scheme to the MSG/SEVIRI satellite images (Schroedter-Homscheidt et al., 2022). The most recent version of the U.S. National Renewable Energy Laboratory's (NREL's) gridded National Solar Radiation Data Base (NSRDB 1998-2016, Sengupta et al., 2018) is also based on a fully physical model. This is the Physical Solar Model (PSM) which was developed by NREL and produces gridded surface solar irradiance estimates using satellite retrievals for clouds and other atmospheric properties from GOES data as input to the radiative transfer model.

The continued developments and improvements since 80s in satellite estimates of surface solar radiation resulted to accurate climatological and real time datasets (Qu et al., 2014; Urraca et al., 2017; Pfeifroth et al., 2023b; Qu et al., 2017; Schroedter-

Homscheidt et al., 2022; Sengupta et al., 2018; Habte et al., 2017) although certain sources of biases and common factors that increase the uncertainty have been reported: the increase of the distance from the subsatellite point, the more frequent occurrence of clouds (especially fragmented cloud cover), complex terrain and bright surfaces (snow, desert). In addition, it is a challenge per se and increases the evaluation uncertainties when any model is validated at an instantaneous time scale. Gridded satellite-estimates with ground-based point measurements of surface solar radiation differ not only due to model uncertainties but also due to different spatio-temporal scales involved (satellite pixels representative of a large area and large time intervals of few minutes, ground-based measurements representative of the area exactly over the station and for smaller time intervals).

Motivated by the recent advances in satellite-based retrievals of surface solar radiation and building upon the knowledge of the already existing and well-established methodologies, an upgrade has been performed to an existing service that provides satellite estimates of surface solar radiation in real time, with the aim the improved nowcasting system to be the basis of the new forecasting system. The Solar Energy Nowcasting System (SENSE) was developed under the EU project Geo Cradle, by the Beyond centre of EO research and satellite remote sensing at the National Observatory of Athens, Greece, in collaboration with the Physical and Meteorological Observatory of Davos, of the World Radiation Center, Switzerland (Kosmopoulos et al., 2018). It is a combination of geophysical input parameters from satellite-based and model data sources and a neural network (NN) technique, trained on precalculated surface solar radiation simulations (look up table – LUT) using RTM. It uses the cloud optical thickness (COT) retrievals produced by the Application Facilities Support to Nowcasting and Very Short Range Forecasting (NWC-SAF) algorithm based on the MSG satellite data and aerosol optical depth (AOD) forecasts from the Copernicus Atmospheric Monitoring Service (CAMS) as inputs to the NN to derive the surface solar radiation in real time. More details about the previous version of the SENSE service can be found in Kosmopoulos et al. (2018). In the same publication the validation of this method showed a good agreement on daily and monthly levels; however, various sources of uncertainties have been identified, concerning mainly the use of the NN especially under high irradiance values, the COT input, and the structure/density of atmospheric parameters in the LUTs. The reason for the development of the new version of the model, called SENSE2, that has been used in the present study, was to minimize those uncertainties, before use it for the new forecasting system. For the new version of the model, it was decided to retain the fully physical approach of the model that benefits from the MSG satellites cloud optical properties monitoring and recent advances in EO and improve the scheme that replace the direct radiative transfer calculations.

The solar energy forecasting methods are categorized into three base methods (Sengupta et al., 2021; Yang et al., 2022) with the time horizon (few seconds to few days) and the exogenous data, i.e., sky cameras, satellite data and numerical weather predictions (NWP). Additionally, there are many statistical and machine-learning methods, which are often combined with NWP data to improve their outputs (post-processing or blending). Each method fits the specific needs of different applications. The use of cloud motion vector (CMV) technique on satellite data is commonly used for solar forecasting of few hours ahead (up to 6 h). Using consecutive satellite images, the CMVs are calculated, and assuming constant cloud speed, the future cloud positions are derived by applying the CMV field to the latest cloud image. The early stages of the use of CMV for short term

forecasting of surface solar radiation based on satellite data start almost twenty years ago (Hammer et al. 1999; Hammer et al. 2003; Lorenz et al. 2004). In the last decade, the interest in using optical flow techniques from the computer vision community in satellite images for cloud motion estimation in the context of solar forecasting has been increased. One of the first works was by Urbich et al. (2018), where for the European domain two optical flow methods were used and compared in forecasting

MSG satellite derived effective cloud albedo. Those forecasted values of effective cloud albedo combined with SPECMAGIC NOW delivers surface solar irradiance short-term forecasts (SESORA -seamless solar radiation, Urbich et al., 2019). An optical flow method to effective cloud albedo maps based on SEVIRI images used by Kallio-Myers et al. (2020) to forecasted global horizontal irradiance up to 4 h ahead with 15 min time resolution for southern Finland, by applying the Heliosat method to forecasted effective cloud albedo maps in combination with the Pvlib Solis clear sky model (Solis-Heliosat forecasting model).

In the same study, they also found that their forecasting model mostly outperforms persistence, especially under changes in cloudiness. It is a common practice to benchmark forecasts of surface solar radiation with the persistence approach (e.g. Kallio-Myers et al.,2020; Garniwa et al., 2023), a method that assumes constant cloudy conditions for the future time steps.

The NextSENSE system was first introduced in the study of Kosmopoulos et al. (2020), as a novel short-term solar energy forecasting system (3 h ahead, every 15 min), based on forecasts of satellite derived COT using a CMV technique, with solar

irradiance estimated by the SENSE model. The NextSENSE system was developed as a continuation of SENSE, during the EU project E-shape and by the same research groups previously mentioned. The employed CMV technique is based on state-of-the-art image processing technologies (dense optical flow). The evaluation of the CMV forecasts was performed by Kosmopoulos et al. (2020) for selected test days with different cloud movement patterns, against the real MSG COT and in term of irradiance estimates using both forecasted and real COT. They found that the deviations of forecasted irradiances

compared with nowcasting outputs ranged from 18% to 34% under changing cloudy conditions, outperforming the persistence method for certain conditions. The aim of the current study is to validate the NextSENSE model for one full year of irradiance forecasts with ground-based measurements, for more robust conclusions. Additionally, as NextSENSE is based on the same hierarchy of SENSE with only addition the CMV analysis, all improvements of SENSE2 are inherited in the new NextSENSE2 system too.

The present study aims to investigate the role of clouds and aerosols in nowcasting and short-term forecasting of global horizontal irradiance (GHI), using ground-based measurements, by:

- Introducing the SENSE2 and NextSENSE2 upgrades of SENSE and NextSENSE systems, respectively.
- Validating the improved nowcasted GHI using ground based pyranometer measurements for 1 year (2017).
- Investigating cloud and aerosol effects on GHI estimates.

- Proposing a possible correction for GHI estimation based on MSG COT real time information.
- Validating CMV forecasted GHI and benchmarking the results with those by the persistence method.

**2 Data and methods**

**2.1 SENSE2**

SENSE2 is an operational system that produces fast estimates of GHI in real time every 15 min, for a wide area including Europe and Middle East-North Africa (MENA) region at high spatial resolution (~5 km), calculated from earth observation (EO) data and look-up-tables (LUTs) derived from radiative transfer model (RTM) simulations. The SENSE2 presented in this study (Fig. 1) is an improved system, compared to the previous SENSE version, in terms of the parameterizations for radiative transfer calculations and, mainly, the improvement of aerosol and cloud representation in the model using a more detailed LUT and multiparametric equations for different aerosol and cloud scenes, respectively. The new version of the SENSE2 system is available as a webservice via https://solar.beyond-eocenter.eu/#solar_short (last access: 2023-12-15).

The first improvements of the SENSE2 system are that:

- the computations of clear-sky GHI are performed in the previous day, for the whole domain (1.5M pixels), every 15 min and, for the current day, the real-time cloud information is applied to provide all skies GHI in real time (no NN is used).
- the computations of clear-sky GHI are based on a new, more detailed LUT of ~16M combinations of simulated GHI at the earth's surface, that was generated using the GRNET High Performance Computing Services and the computational resources of ARIS/GRNET infrastructure. The RTM simulations were performed using the libRadtran package (Emde et al., 2016; Mayer & Kylling, 2005) and Table 1 summarizes the input variables and their resolution resulted to the ~16M runs.

RTM simulations were performed spectrally from 280 to 3000 nm, with 1 nm spectral resolution using the DISORT radiative transfer solver in pseudo-spherical mode (Buras et al., 2011). The molecular absorption parameterization of representative wavelength approach – REPTRAN was used in the solar range (Gasteiger et al., 2014) to account for the absorption of atmospheric gases for the whole solar spectrum. The Kurudz 1.0 nm (Kurucz, 1994) extraterrestrial solar spectrum and the U.S. Standard Atmosphere (Anderson et al., 1986) were used as inputs. The default aerosol model of Shettle (1989) was used as the basis with modified aerosol optical properties of AOD, single scattering albedo (SSA) and Ångström exponent (AE) varied according to Table 1. The spectral global irradiances were integrated over the spectral range of the simulations to derive the GHI.

The clear-sky GHI estimates by SENSE2 (Fig. 1) are calculated in the previous day, by linear interpolation in the 7 dimensions (7D) of the precalculated GHI LUT using the corresponding inputs. Specifically, the solar zenith angle (SZA) values are precalculated for every grid cell of the domain (1.5M in total), for the 15 min time step. The main input parameter for the clear-sky computations is the forecasted AOD at 550nm from CAMS (CAMS AOD hereafter). The forecasts for the day of interest are values from the CAMS run of the previous day initialized at 00:00 UTC (e.g., the AOD used to simulate the GHI for the 24[th] of a month, has been derived from the CAMS run that started on the 23[rd] at 00:00 UTC). Climatological values are used for the interpolation in the 7D LUT for the additional aerosol optical properties of SSA and AE (MAcv2 climatology, Kinne,

2019), for the water vapor (WV) (CAMS reanalysis (Inness et al., 2019)), the total ozone column (TOC) (OMI TOC data (Bhartia, 2012) based climatology) and surface albedo (GOME-2 database of directionally dependent Lambertian-equivalent reflectivity (Tilstra et al., 2017, 2021)). It should be mentioned that the interpolation procedure in the 7D LUT was added in the new SENSE2 to further improve the accuracy of the GHI estimations. Finally, since the results of the RTM runs are at sea level and for the mean earth-sun distance, a post correction of the clear-sky GHI values from the LUT is performed for the

surface elevation following the methodology described in Fountoulakis et al., (2021) and the actual earth-sun distance for the particular day of the year (DOY). Based on simulations for various atmospheric and surface albedo conditions, Fountoulakis et al. (2021) estimated an average increase of the GHI by 2% per km, which has been also applied to the model output to correct the surface GHI for sites at higher altitudes than sea level.

The use of LUTs at operational surface solar radiation retrievals instead of direct RTM calculations is well established (e.g.,

Qu et al. 2017; Mueller et al., 2009). From a technical point of view, various concepts exist which can reduce by several orders of magnitude the number of RTM simulations needed for a LUT generation. Mueller et al. (2009) developed a flexible, fast, and accurate scheme to retrieve the broadband surface solar irradiance (CM-CAF datasets) using the hybrid eigenvector approach, resulting in a combination of basis LUTs with optimized interpolation grid and parameterizations, using only almost one thousand RTM calculations. This approach was extended by Mueller et al., (2012) to wavelength bands for spectrally

resolved surface solar irradiance retrievals from spaceborne data. This optimization of the computing performance is of paramount importance for the reprocessing of a large amount of satellite data (up to a few decades). In this work, the main concept behind the generation of our clear sky LUT was to have spectral irradiance outputs (1nm spectral resolution). The choice to calculate spectral solar data and not directly shortwave radiation is based on the fact that the SENSE2 output could be used for other (health, agriculture, other) applications, based on the irradiance weighting of a relevant spectral range with

an action spectrum (function) defined for each of the effects. So, a large number of RTM runs had to be performed (once) for spectral surface solar irradiance covering all possible combinations of atmospheric and surface states. Technically, since the operational set up of the SENSE2 model allows for the computation of the clear sky GHI values from the previous day, the processing time for interpolation to the 7 dimensions of the LUT has no effect in producing timely the real time output of the model every 15 min, while the accuracy of the clear sky output is almost identical with direct RTM simulations

(Papachristopoulou et al., 2022) and the uncertainties of the clear sky GHI retrievals are related only to the uncertainties of the model inputs. In addition, this LUT includes various aspects especially for aerosols (AOD, SSA, AE) that can reduce the uncertainty under different aerosol conditions, for broadband solar radiation or specific spectral regions.

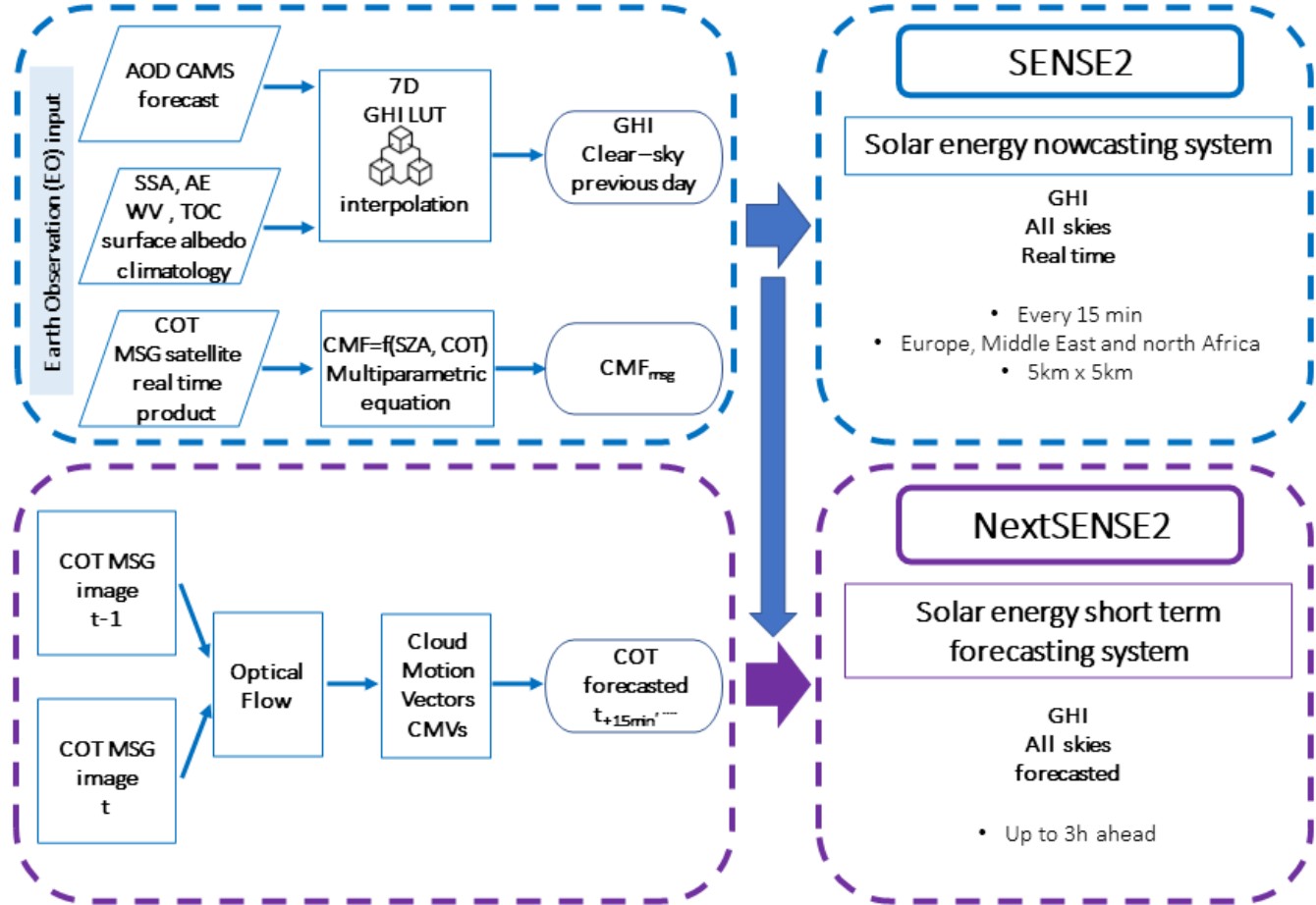

**Figure 1 Schematic overview of solar energy nowcasting system (SENSE2) and short-term forecasting (NextSENSE2) up to 3h ahaid.**

**Table 1 Input parameters to radiative transfer simulations performed at the ARIS GRNET supercomputer resulted to the 7D GHI LUT.**

| Parameter | Range | Resolution |
|---|---|---|
| Solar zenith angle (SZA in deg) | 1 to 89 | 1 |
| Aerosol optical depth at 550nm (AOD) | 0 to 2, 2.5,3.0 | 0.05 |
| Single Scattering Albedo (SSA) | 0.6 to 1 | 0.1 |
| Angstrom exponent (AE) | 0 to 2 | 0.4 |
| Total Ozone Column (TOC in DU) | 200 to 500 | 100 |
| Water Vapor (WV in cm) | 0.5 to 3 | 0.5 |
| Surface albedo | 0.05 to 0.8 | 0.15 |

Another improvement is related to the cloud representation in real time, using multi-parametric equations for different cloud scenes, based on the Cloud Modification Factor (CMF) concept, instead of using the COT as an input parameter in direct RTM calculations. The computation of the all-skies GHI in real time, every 15 min, is based on the COT product we extract operationally in real time using broadcasted MSG satellite data and the software package provided by EUMETSAT Satellite Application Facilities of Nowcasting and Very Short Range Forecasting, NWC SAF (Meteo France, 2016; Derrien & LeGLeau, 2005). To provide timely the all skies GHI SENSE2 product for 1.5M pixels, neither the direct radiative transfer simulations nor the multi-dimension interpolations would be sufficiently fast. Instead, a multi-parametric equation was constructed, fitted on libRadtran simulations for a wide range of COT values for different SZAs (points in Fig.2a). The design of the cloud model was a trade -off between the relevance of the cloud property and the operational implementation of the model. It has been shown in previous studies (Qu et al. 2017) that for most of the cases (except for high surface albedo values >0.9), the cloud vertical position and extent has a small or negligible influence for the RTM simulations of surface solar irradiance. Under cloudy conditions, COT is the variable that has the greatest impact on simulating surface solar radiation (Qu et al., 2017, Oumbe et al., 2014, Taylor et al., 2016). In our simulations, spherical droplets were assumed, with typical values for the effective radius (Reff = 10 μm) and typical climatological mean heights (base at 2 km, 3 km height) (Taylor et al. 2016, Kosmopoulos et al., 2018), given the unavailability of height descriptors in the operational mode and the negligible influence of changes in droplet effective radius with respect to COT on simulating surface solar radiation (Oumbe, 2009) and towards simplify the cloud model. The COT of the cloud layer is additionally specified at 550 nm, which leads to an adjustment of the liquid water content default value of 1 $g/cm^3$, using the parameterization by Hu and Stamnes (1993). Finally, for the libRadtran simulations homogeneous layer clouds were used, meaning cloud cover fraction value of 100%, which is one of model limitations, since assuming totally cloudy pixels is not always correct for low values of COT (Mueller et al. 2009). The simulated GHI for each COT was divided by the GHI for COT=0 (clear sky) for the same SZA to derive the CMF (Eq. 1). The CMF ranges from zero (overcast conditions) to 1 (clear sky) and it is easy to use to provide all skies GHI by simply multiplying clear-sky GHI with CMF (Eq. 3). The libRadtran-derived CMF for each SZA were fitted against COT using the hyperbolic tangent function. The resulting fits are shown as solid lines in Fig.2a and are mathematically expressed by the multi-parametric Eq. 2.

$$CMF = \frac{GHI}{GHI_{clr}} \qquad (1)$$

$$CMF = 1 - \tanh^b(COT^a) \qquad (2)$$

where a and b are polynomials of SZA

$$a = 2.24 \cdot 10^{-1} + 2.81 \cdot 10^{-4} \cdot SZA - 2.18 \cdot 10^{-5} \cdot SZA^2 + 3.71 \cdot 10^{-7} \cdot SZA^3 - 2.65 \cdot 10^{-9} \cdot SZA^4 \qquad (2a)$$

$$b = 12.2 + 5.27 \cdot 10^{-3} \cdot SZA - 2.24 \cdot 10^{-3} \cdot SZA^2 + 8.33 \cdot 10^{-6} \cdot SZA^3 + 3.94 \cdot 10^{-8} \cdot SZA^4 \qquad (2b)$$

The real time MSG COT is used as input in Eq. 2 every 15 min, along with SZA, for ~1.5M pixels, to calculate the CMF (CMFmsg hereafter). Apart from being very fast, the use of this formula to calculate CMFmsg is also accurate, as it can be seen by the comparison of the CMF values derived by Eq. 2 against those of libRadtran runs (Fig. 2b). CMF differences are

less than 0.015 (or 1.5%) for SZA lower than 70 degrees, while they are up to 0.03 (3%) for SZAs between 80 and 90 degrees, showing the very good representation of the CMF as a function of COT with Eq. 2. In terms of accuracy this means that using Eq. 2 is almost the same as running RTM simulations, but in terms of computational time is by far more efficient in the operational mode. Finally, by multiplying CMFmsg with clear-sky GHI, the all-skies GHI product is provided (Eq. 3), in less than 1 min for 1.5M pixels.

$$GHI = GHI_{clr} * CMFmsg \qquad (3)$$

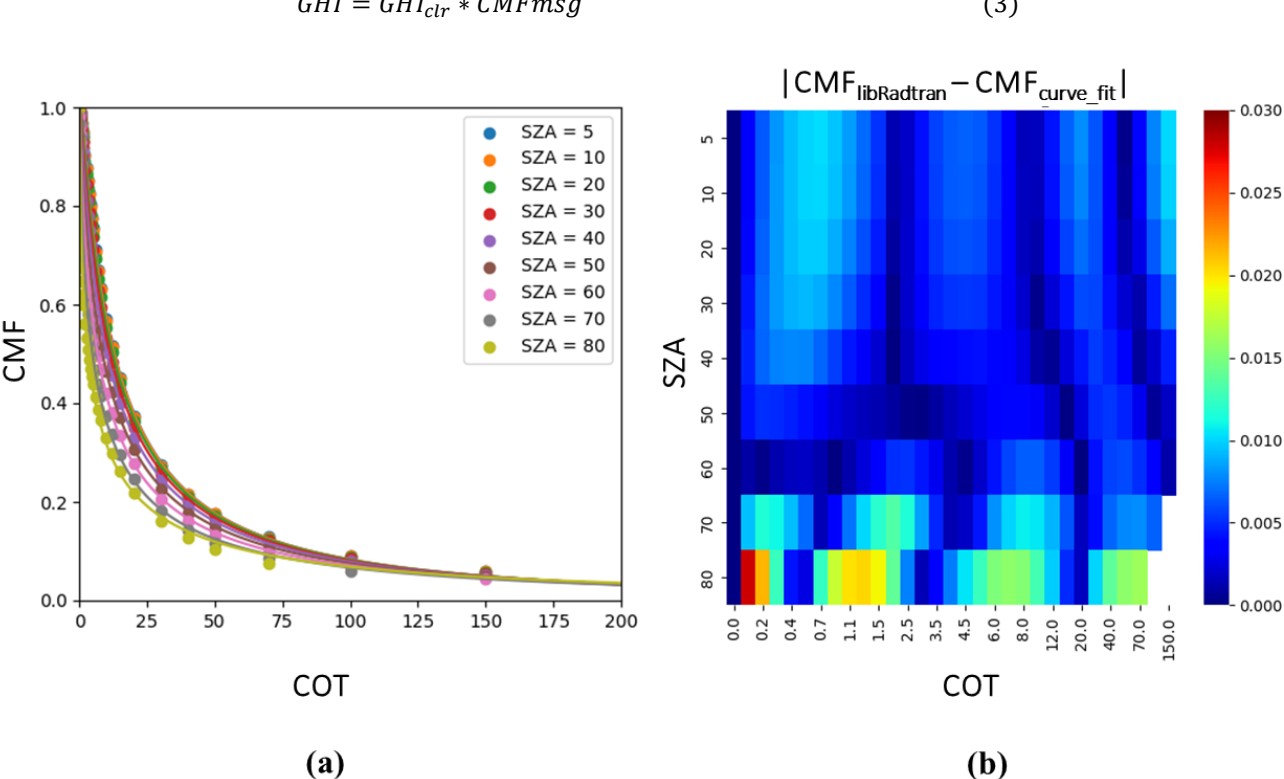

**(a)**                                                                **(b)**

**Figure 2 (a) Cloud modification factor (CMF) versus cloud optical thickness (COT) and solar zenith angle (SZA) based on radiative transfer simulations of global irradiances using the libRadtran package. CMF is the ratio of global horizontal irradiance (GHI) values to those under cloudless conditions (COT=0). (b) Differences between the CMF directly derived from libRadtran simulations against those derived from Eq. 2, as a function of COT and SZA.**

The new SENSE2 configuration was built for the improvement GHI nowcasting on the 15 min time scale. Additionally, it allows a greater flexibility for the system to:

- include reanalysis or measured data of AOD and other optical properties e.g., CAMS reanalysis or AERONET measurements.
- extent to other output products. Apart from GHI, direct normal irradiance (DNI) or total irradiance on tilted surface could be also produced. By introducing spectral information and making the appropriate modifications, products related to specific spectral regions could be also derived (e.g., UV Index by using real time input for TOC, photosynthetically active radiation (PAR) etc).

- run a past time series of one or few locations, autonomously, using as input actual measurements. In this case, if there is no time constrain, model runs could be performed without the parametrizations (LUT and multiparametric functions).

## 2.2 NextSENSE2

NextSENSE2 is the operational system that provides forecasts of GHI up to 3-hours ahead with a 15-min time step by applying a CMV technique to the MSG COT product (Fig. 1). In this section, we describe the method employed to produce forecasted COT, which is the main input to derive the operational forecasts of GHI. All the other EO inputs and the radiative transfer parameterizations for fast estimates of forecasted GHI are the same as those described in the previous section for the SENSE2 model.

We use CMVs to predict the motions of the clouds and project their future positions. The CMVs in NextSENSE2 are calculated
by applying a state-of-the-art optical flow algorithm from the computer vision community. Optical flow is the apparent motion of objects between consecutive frames, caused by the relative movement between the object and a camera. We apply the Farnebäck (2003) two-frame motion estimation technique to images of COT product (Kosmopoulos et al., 2020). Several other optical flow algorithms like TV-L1 are available as free software (OpenCV) and are used for cloud motion estimation in solar energy short-term forecasting systems (Urbich et al., 2019). In this study we used Farnebäck based on the results of the previous
study by Kosmopoulos et al. (2020). By applying the algorithm to two consecutive images of satellite derived COT the optical flow displacement vectors are calculated. This CMV field is applied to the later COT image (real) to get the next COT image (forecasted COT). This procedure is repeated for 12 times resulting to the 3h forecasting horizon. Main assumptions are the brightness constancy and that the cloud's displacements are only two dimensional (image plane). More details regarding the CMV model and forecasted COT can be found in Kosmopoulos et al. (2020).

## 2.3 Persistence forecast

It is not easy to evaluate the quality of different forecasting methods of surface solar radiation using only statistical metrics, since the study period, the geographical area and other factors are affecting their forecasting accuracy. That's why it is a typical practice of evaluation to benchmark the different forecasts against some simple forecast methods (Pelland et al., 2013). We used the persistence forecast to benchmark the CMV forecasted GHI of NextSENSE2 system which is a commonly used
reference in solar forecasting (e.g. Kosmopoulos et al., 2020; Kallio-Myers et al., 2020). This method assumes that the state of the clouds remains constant for future time steps, while all other variables like SZA etc. dynamically change. Hence, it uses the same COT values from the later satellite information as input to the next 12 time-steps in order to forecast GHI up to 3h ahead.

## 2.4 Ground based irradiance measurements

To validate the modelled GHI, ground-based measurements from pyranometers were utilized. The 1-min GHI ground based measurements were collected from stations of the Baseline Surface Radiation Network (BSRN; Driemel et al., 2018) that are

within the study area and have data throughout 2017, and from two additional stations at Athens (ASNOA: NOA's Actinometric Station) and Thessaloniki. Table 2 summarizes the information of the 10 in total stations utilized and Fig. 3 depicts their geographical locations.

BSRN station-to-archive files were accessed and manipulated using the SolarData v1.1 R package (Yang, 2019). The function that reads the data from the station-to-archive files also computes several auxiliary variables such as solar zenith angle, clear sky irradiances using the Ineichen-Perez clear sky model (Ineichen & Perez, 2002) and extraterrestrial GHI. Using the same methodology, the Ineichen-Perez clear sky model values were also computed for the non BSRN station data, by adjusting the functions of the SolarData v1.1 R package for the non-BSRN stations.

The BSRN recommended Quality Check (QC) tests (Long & Dutton, 2010) were performed to the collected measurements, to ensure the best quality of the measurements. Measurements that were not respecting the above QC tests, were flagged, and set to as a missing value. The GHI records that are available at the two Greek stations (1951 – present in Athens, 1993 – present in Thessaloniki) are among the longest continuous high quality GHI records at the Eastern Mediterranean Basin, an area where BSRN data are not available for the period of this study. The pyranometers in Athens and Thessaloniki are calibrated regularly

and the GHI measurements have been subjected to quality control before being used in the study. More information for the GHI datasets at the two stations can be found in Bais et al., (2013) for Thessaloniki, and Kazadzis et al, (2018) for Athens.

**Table 2 Detailed information about the ground-based sations used in this study.**

| Name | Ground based pyranometer | | | | AERONET station |
| --- | --- | --- | --- | --- | --- |
| | network | Lat. (°N) | Lon. (°E) | location | |
| ATH - Athens | - | 37.9 | 23.7 | Europe/Greece | Co – located |
| CAB – Cabauw | **BSRN** | 51.9711 | 4.9267 | Europe/Amsterdam | Co – located |
| CAM – Camborne | **BSRN** | 50.2167 | -5.3167 | Europe/London | Co – located |
| CAR – Carpentras | **BSRN** | 44.083 | 5.059 | Europe/Paris | Co – located |
| CNR – Cener | **BSRN** | 42.816 | -1.601 | Europe/Madrid | Co – located |
| LER – Lerwick | **BSRN** | 60.1389 | -1.1847 | Europe/London | Co – located |
| LIN – Lindenberg | **BSRN** | 52.21 | 14.122 | Europe/Berlin | Co – located (metObs LIN) |
| PAL – Palaiseau, SIRTA Obser | **BSRN** | 48.713 | 2.208 | Europe/Paris | Co – located |
| TAM – Tamanrasset | **BSRN** | 22.7903 | 5.5292 | Africa/Algiers | Co – located |
| THE -Thessaloniki | - | 40.63 | 22.96 | Europe/Greece | Co – located |

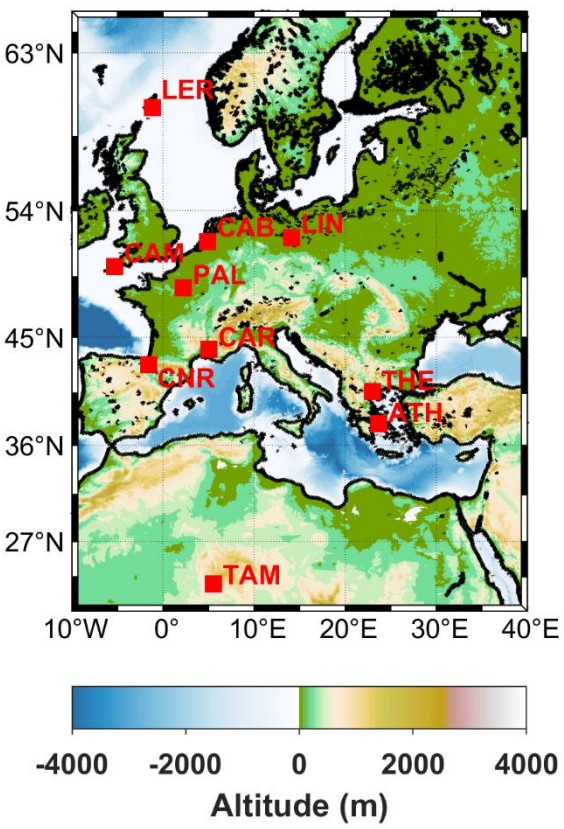

**Figure 3 Locations of the ground-based stations measuring global horizontal irradiance (GHI) that are used in the current study. These are eight BSRN stations, plus Athens, and Thessaloniki, Greece.**

### 2.5 Ground based aerosol information

To assess the CAMS AOD forecasts used as input to the model, ground-based measurements of AOD from the AERONET network (Holben et al., 1998) were used. All the ground-based stations with pyranometer data (BSRN, Athens and Thessaloniki) have a collocated AERONET station (see Table 2). The level 2, version 3 direct sun (Giles et al., 2019) AOD data at 500nm were collected and using the Ångström exponent for 440-675nm, the AOD values at 550nm were derived. Only for Cabauw, measurements of AOD at 500 nm weren't available, therefore AOD at 440 nm was used instead and converted to 550 nm using the Ångström exponent for 440-675 nm.

### 2.6 Evaluation metrics

For the validation of the SENSE2/NextSENSE2 derived GHI values against ground-based measurements, common statistical metrics have been adopted. Given that the error is defined as the difference between modelled values ($x_{m\_i}$) and observed values ($x_{o\_i}$), we have the following common metrics:

Mean Bias error:

$$MBE = \frac{1}{N}\sum_{i=1}^{N}(x_{m_i} - x_{o_i}) \qquad (4)$$

Root mean square error:

$$RMSE = \sqrt{\frac{1}{N}\sum_{i=1}^{N}(x_{m_i} - x_{o_i})^2} \qquad (5)$$

And Pearson correlation coefficient R. The relative values of those metrics rMBE and rRMSE were obtained with respect to the mean of the observed values of GHI.

An additional metric the forecast skill (FS) was used to assess the performance of CMV forecasted GHI using persistence model as a benchmark model:

$$FS = 1 - \frac{rRMSE_{CMV}}{rRMSE_{pers.}} \qquad (6)$$

where rRMSE$_{CMV}$ and rRMSE$_{pers}$ are the relative RMSE of the CMV and persistence forecasting models, respectively.

## 3 Results and Discussion

The results are discussed separately for the evaluation of nowcasted GHI (Section 3.1) related to SENSE2 outputs (modelled GHI hereafter) and the evaluation of the short-term forecasted GHI (Section 3.2), namely the NextSENSE2 product (forecasted GHI, hereafter). The comparisons between ground based and estimated GHI where restricted to SZAs below 75° (i.e., for solar height above 15° from the local horizon), because for higher SZAs the accuracy of the satellite cloud retrievals degrades. The CMF derived from the ground-based measurements of GHI was used in our analysis to evaluate CMFmsg and to categorize

the cloudiness conditions. Specifically, the CMF was calculated as the ratio (Eq. 7) of measured GHI to the clear sky irradiance calculated by the Ineichen-Perez clear-sky model (Ineichen & Perez, 2002) (See Section 2.4).

$$CMF = \frac{GHI_{measured}}{GHI_{clr}} \qquad (7)$$

Three categories according to CMF are used in the following: CMF≥0.9 for clear sky conditions, 0.4<CMF<0.9 for partially cloudy conditions and CMF≤0.4 for overcast conditions.

### 3.1 Nowcasting

### 3.1.1 Overall performance

Figure 4 presents the overall performance of the SENSE2 system at the (instantaneous) 15 min time scale, by comparing the modelled GHI values against ground-based measurements, for all stations, for a whole year (2017). We can see that most of

the points (Fig. 4a, number of cases N>600) fall on the 1:1 line (blue line) which indicates the overall good performance of the system, with a correlation coefficient of 0.93. For 58 % of the cases, the absolute differences between modelled and ground-based measurements of GHI are within +/-50 W/m² or +/-10 % (Fig. 4b). The SENSE2 system mostly overestimates the GHI, which corresponds to points above identity line (Fig. 4a), with MBE 23.8 W/m² (4.9%), which is more pronounced for low irradiances (low left corner of Fig. 4a with GHI<250 W/m²) Lerwick is the most northern station and at the same time the station with the greatest MBE (Fig. 4b and Fig A1,2 in Appendix A).

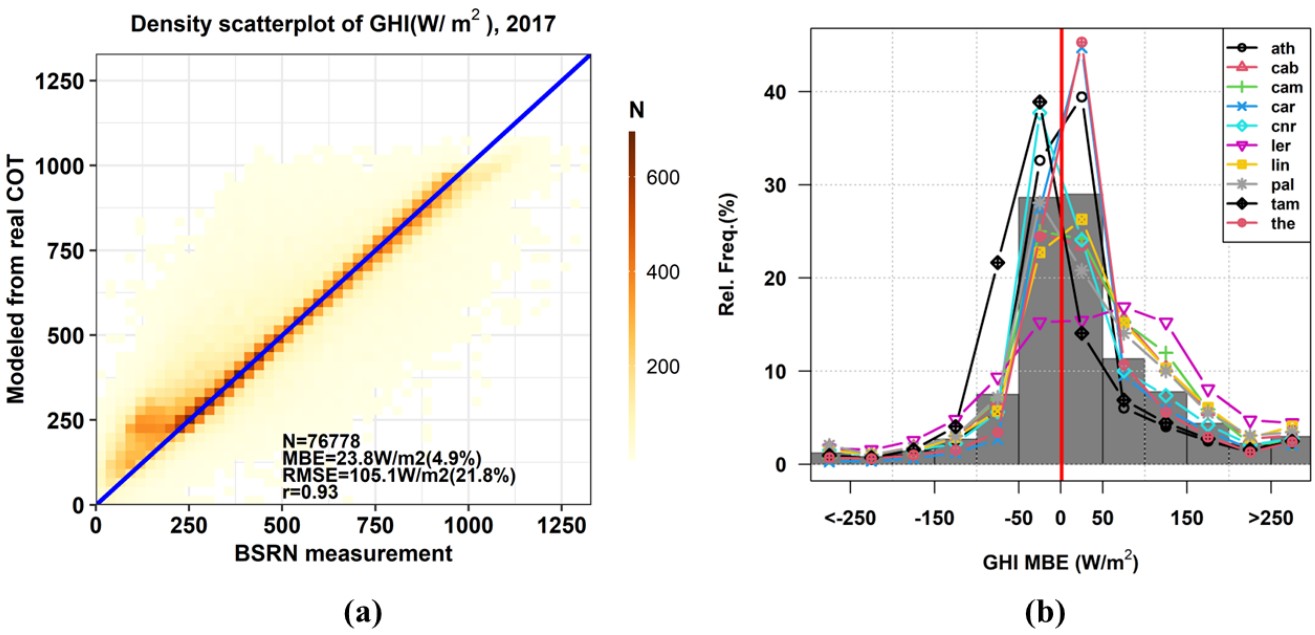

**(a)**  **(b)**

**Figure 4 (a) Comparison of the global horizontal irradiance (GHI) modeled versus measured for all ground based stations, for 2017. (b) Relative frequency of GHI MBE for all stations (grey bars) and for each station (lines with different symbols and colors).**

We investigated the influence of the mean cloudiness (CMF) of every station along with its latitude and the mean measured GHI to the GHI MBE and is presented in Fig. 5. The GHI MBE increases with an increase in cloudiness (decrease in mean CMF). At the same time the cloudiness increases with latitude, where lower values of mean measured GHI are observed as well. Those results are in line with previous studies (Qu et al., 2014; Qu et al., 2017). According to Qu et al., (2014) the error of the satellite estimates of surface solar radiation increases with an increase of the distance from the subsatellite point (lat=0°, lon=0° for Meteosat) and in occurrence of fragmented cloud cover. Qu et al (2017) found lower accuracy of their retrievals for the northernmost stations which was attributed to the more frequent cloud occurrence over those stations and the more erroneous satellite retrievals of clouds properties for large SZAs and satellite viewing angles. One of those stations was Lerwick, which is close to the edge of the field of view of Meteosat satellite, where errors due to parallax becomes important

(Marie-Joseph et al., 2013; Schroedter-Homscheidt et al. 2022). The cloud effect on GHI estimates is investigated in more detail in Section 3.1.3.

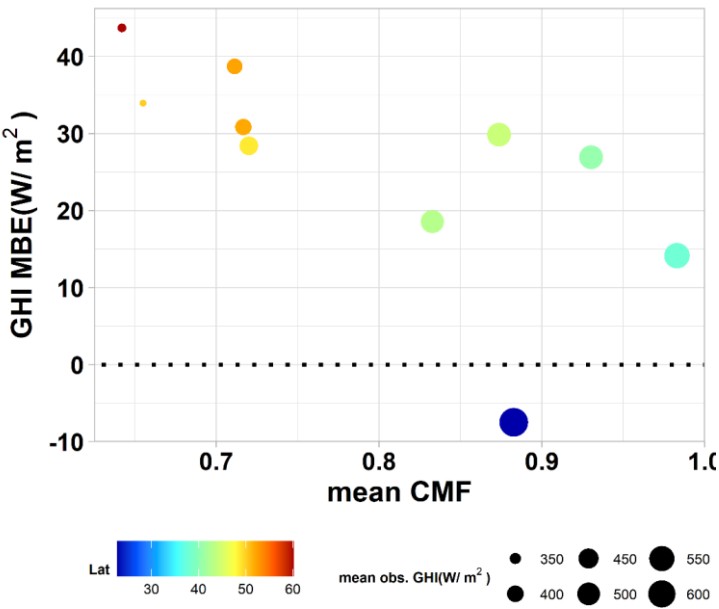

**Figure 5 GHI MBE dependence on the mean CMF for all stations. The colour of the points indicates the latitude of the station and their size the magnitude of the mean GHI observed at the ground-based stations.**

All the statistical metrics are drastically improved with increasing time scale for all stations (Fig. 6). Stations with similar results are the northern most stations CAB, CAM, LER, LIN, and PAL. At the 15 min time scale their MBE, RMSE and correlation coefficient range 29-43 W/m$^2$, 104-131 W/m$^2$ and 0.82-0.90, respectively. Those statistics are improved for the monthly means to 5-10 W/m$^2$ for MBE, to 7-13 W/m$^2$ for RMSE and to r~1. Similar results were found for the rest southernmost stations (MBE, RMSE and correlation coefficient range from -7 to 30 W/m$^2$, from 84 to 104 W/m$^2$ and from 0.93 to 0.95, respectively, for 15 min time scale and from -4 to 8 W/m$^2$, from 6 to 10 W/m$^2$ and r~1, respectively, for monthly means). The overall MBE and RMSE is reduced to 6.6 W/m$^2$ (3.3%) and 15.4 W/m$^2$ (7.7%) for the daily mean GHI and to 5.7 W/m$^2$ (3.2%) and 9.2 W/m$^2$ (5.2%) for the monthly means, while correlation coefficient reaches values up to almost 1, which was anticipated since the cloud effect is smoothed out for larger time scales.

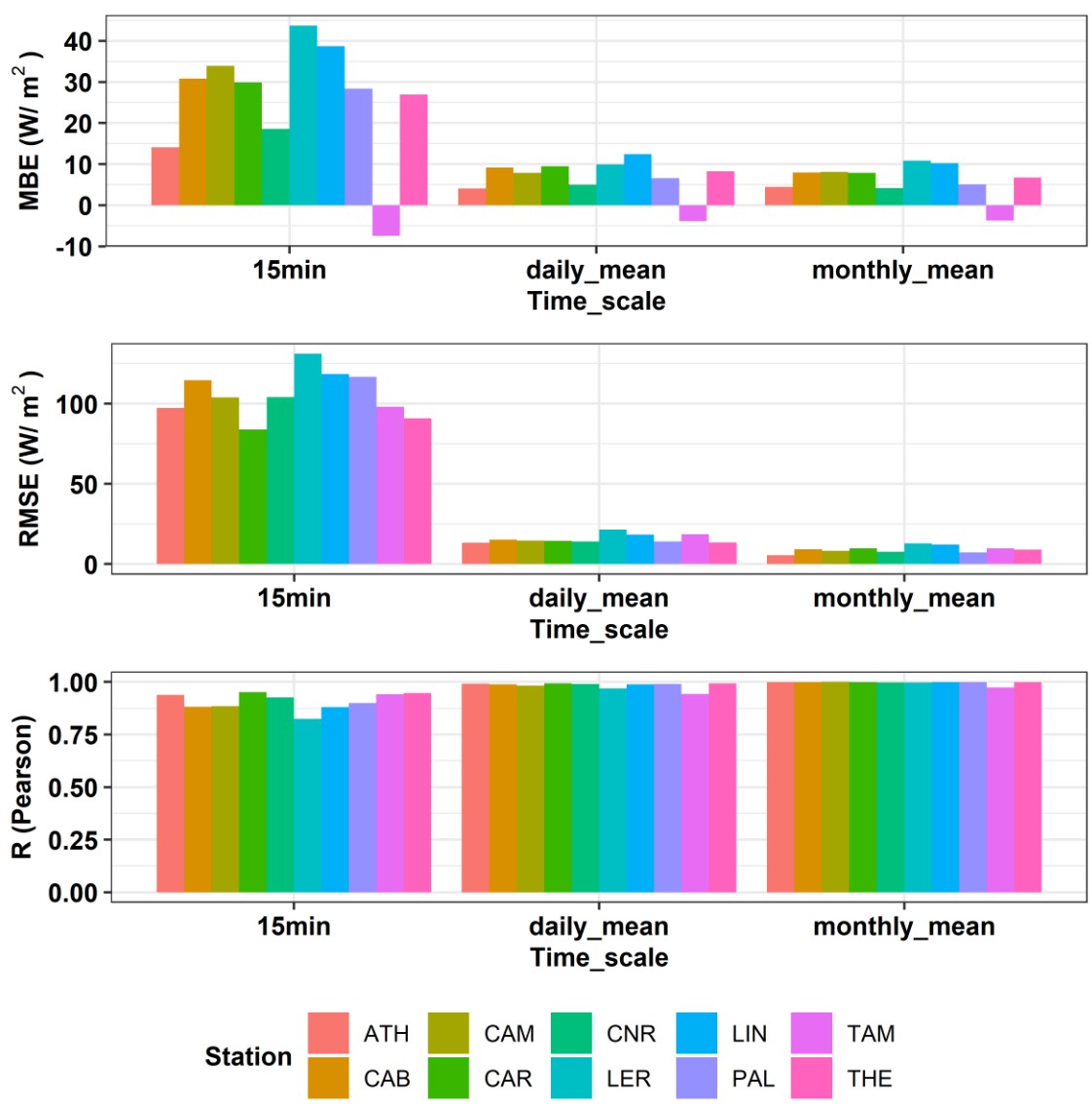

**Figure 6 Comparison of the global horizontal irradiance (GHI) modeled versus measured per ground based station, for 2017, for different time scales (15 min, daily mean and montlhy mean).**

### 3.1.2 Aerosol effect on retrieved solar irradiance

The CAMS AOD forecasts used as input to the operational model were assessed against ground-based measurements from the AERONET network, and the related uncertainty introduced to modelled GHI has been calculated. The AERONET AOD direct sun measurements were matched with CAMS AOD forecasts (1h time resolution) interpolated to the 15 min time steps of the model. The closest AERONET measurement +/- 10 min around the 15 min time steps were matched (or the mean value if more than one measurement were available). To estimate the model uncertainties due to forecasted AOD, the clear sky GHI

was calculated using as input first the forecasted CAMS AOD and second, the synchronized AERONET AOD measurements. The comparison for AOD and modelled GHI is presented in Fig. 7 per station in terms of MBE.

CAMS forecasts mostly overestimate AOD with MBE 0.015 (10%) for all stations which results to an underestimation of modelled clear sky GHI -2.7 W/m$^2$ (-0.4%). The greatest overestimation 0.05 (~50%) was found for CAM and CNR which resulted to the greatest underestimation of clear sky modelled irradiances -8.5 W/m$^2$ (-1.4%). Underestimation of AOD was found for CAB and THE, with MBE<0.01 (<3%), resulting in negligible overestimation of modelled irradiances (MBE< 1W/m$^2$ or 1%).

An overestimation of the CAMS forecasted AOD at 550nm is also reported for 2017 over Europe (average modified normalized mean bias ranging from ~10 to 30%) from the continuous quarterly evaluation of the AOD forecasts against daily AERONET cloud-screened (i.e. Version 3 level 1.5) sun photometer data (Basart et al., 2023; Eskes et al., 2021). While this is the case on average, in contrast during high aerosol loads, CAMS forecasted AOD is underestimated, especially in desert regions and during dust events (Basart et al., 2023; Papachristopoulou et al., 2022) which might explain the almost zero bias for Tamanrasset station (the overestimation of small AODs masked out by the frequent underestimation of large AODs) compared to the greater values of bias (>0.01) found for most of the rest stations. Qu et al. (2017) analysed case studies at Tamanrasset and found that the CAMS (MACC) AOD at 550nm is frequently underestimated against AERONET data during summer dust events, explaining the strong positive bias they found for their modelled direct irradiance (using Heliosat-4 method and the McClear clear sky model). In contrast to the CAMS AOD underestimation during dust events, in the same study (Qu et al., 2017) a systematic overestimation of AOD was found during periods free of those events for the two examined desert stations (Sede Boqer and Tamanrasset), to which they associated the underestimation of their modelled direct irradiance.

The updated McClear v3 clear sky model used in study by Schroedter-Homscheidt et al. (2022) and for their GHI estimates under clear-sky conditions a negative bias was found for most of the station especially for those located in dust affected regions, which is in line to our results although not directly comparable since they compared directly with the BRSN measured irradiances. Our results demonstrate the good performance of the clear sky model using CAMS forecasts, highlighting that AOD product forecasted by CAMS is suitable for GHI nowcasting applications.

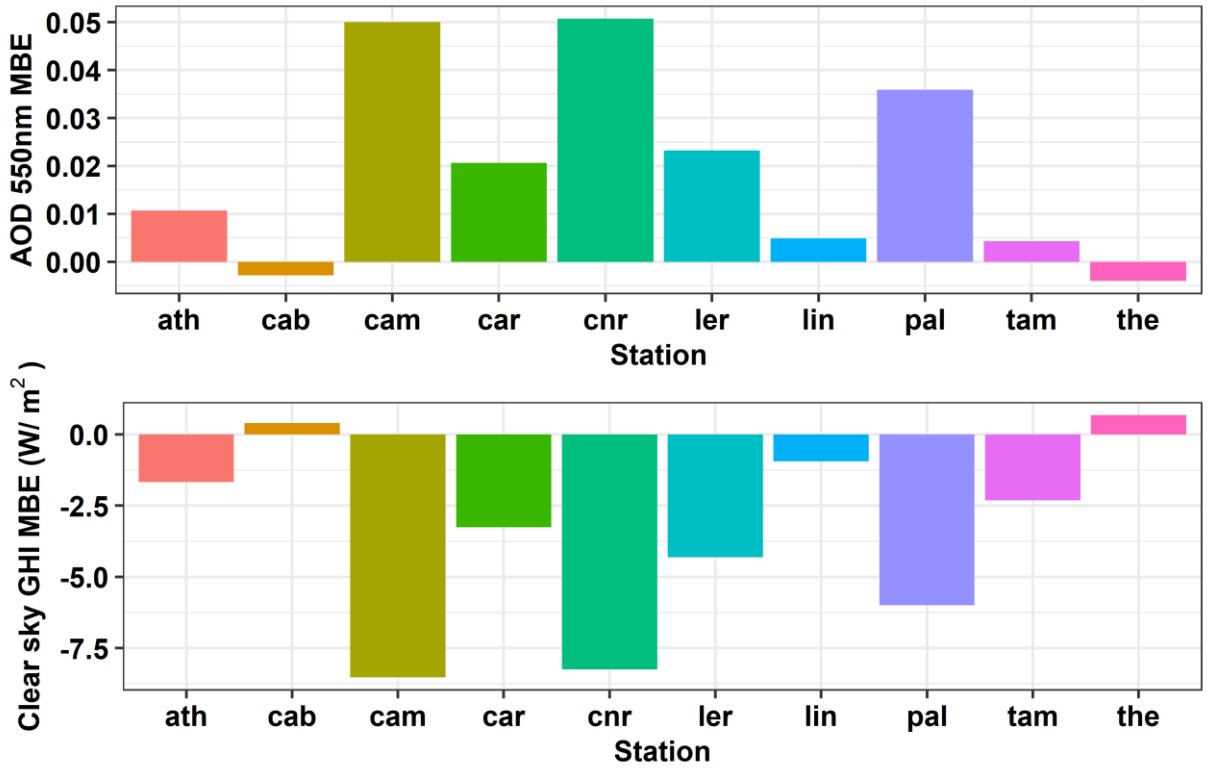

**Figure 7 Upper panel: Mean bias error (MBE) of aerosol optical depth (AOD) at 550nm forecasted by CAMS (1 day ahead forecast) compared to AOD measured by ground based sun photometers of the AERONET network. Lower panel: MBE of global horizontal irradiance (GHI) modeled under clear sky conditions using as input CAMS forecasted AOD at 550nm versus measured values (AERONET).**

### 3.1.3 Cloud effects on retrieved solar irradiance

Overall, the model relatively overestimates GHI, as we saw in Section 3.1.1.The improvement of the statistics going from instantaneous comparison to integrated time scales (e.g. daily) points to the direction that such overestimation can be attributed both to the uncertainties related to the cloud information from satellite retrievals, but also to satellite/ground-based evaluation representativity issues. In order to investigate this more, we decomposed the error in modelled GHI for different conditions in cloudiness.

Initially, we classified the cloudiness conditions using ground-based CMF (Fig 8a, b, c). According to the results, GHI is overestimated by the model under cloudy conditions (CMF<0.9), while for clear sky conditions (CMF≥0.9, Fig. 8a) the model closely resembles the measured GHI. For partially cloudy conditions (0.4 < CMF <0.9, Fig. 8b), the MBE is 81.6 W/m$^2$ (22.8%) and the greatest error in GHI occurs for the low CMF values (CMF≤0.4, Fig. 8c) (MBE=100.1 W/m$^2$ or 73.1%). In the latest category, the largest occurrence of high deviations at low measured GHI values (<250 W/m$^2$) is found.

We compared also the modelled and measured GHI values for clear sky conditions according to the satellite data, namely COT=0 (Fig. 8d). In this case, the model overestimates GHI with MBE 13.6 W/m$^2$ (2.3%). Most of the cases are on the 1:1 line, with few ones being higher , especially, for measured GHI<250 W/m$^2$,meaning that there are clouds over the ground-based station that haven't been resolved by the satellite pixel (COT=0). A positive bias was also found for all stations examined by Qu et al. (2017) for cases of clear sky pixels as defined by APOLLO/SEV cloud properties retrieval scheme which contributes to the overall overestimation for the all sky conditions, and it was attributed to small broken clouds causing large variability in surface GHI, and due to a false detection by the cloud retrieval algorithm being treated as clear sky cases.

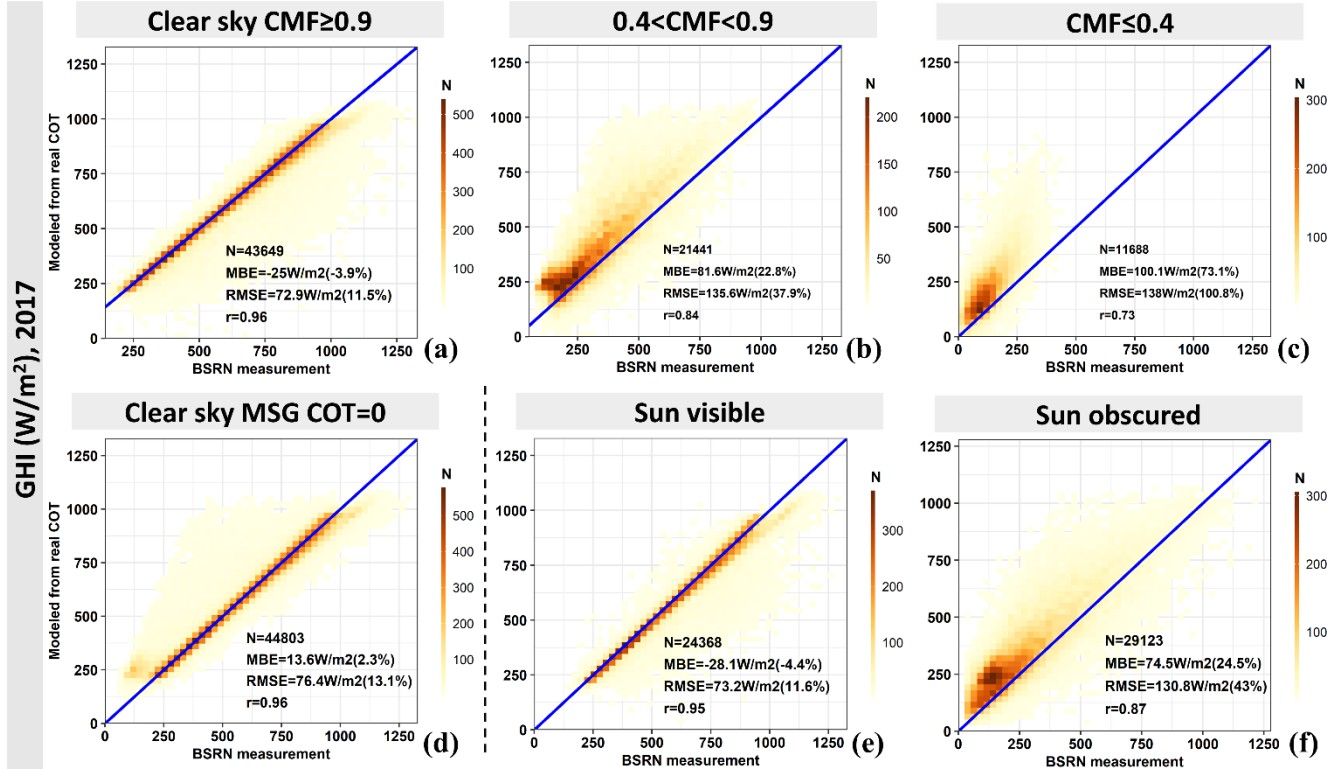

**Figure 8 Comparison of the global horizontal irradiance (GHI) measured versus modeled for all ground based stations (a, b, c) for different cloudiness conditions based on the cloud modification factor (CMF) derived by the ratio of GHI ground based measurements devided by clear-sky GHI (clear-sky model) (d) for clear-sky conditions as determined by MSG satellite product by zero values of cloud optical thickness (COT=0) (e, f) for conditions characterized as sun visible or obscured.**

To demonstrate the effect of sun visibility over the ground-based stations to the GHI error, we tried to separate those instances by using the pyrheliometer measurements of direct irradiance (DNI – direct normal irradiance) available by the BSRN network. The DNI measurements (1 min) were divided with the clear-sky DNI, again from the Ineichen-Perez clear sky model (Ineichen & Perez, 2002). We classified as sun visible the situations with ratio of actual to clear sky model DNI >0.8. This threshold was selected to account for the strong effect of aerosols in DNI, given that a monthly mean climatological value for aerosol attenuation factor is used by DNI clear-sky model (Ineichen & Perez, 2002). We classified as sun obscured situations with ratio <0.6, and we omitted as unclassified situations those with ratio values between 0.6 and 0.8, to be confident that direct

irradiance is blocked by clouds by more than 40%. The results of the GHI comparison between modeled and measured values were grouped based on the sun visibility classification and are presented in Fig. 8e,f. We can see that the sun visible situations give quite good results (points close to the 1:1 line, with MBE -28.1 W/m$^2$ or -4.4%).

In contrast, the model overestimates GHI (MBE 74.5 W/m$^2$ or 24.5%) when the sun is obscured over the ground-based station. Comparing Fig. 8b and Fig. 8c with Fig 8f we can see that most of the above the 1:1 line cases happen when the sun is obscured. This is caused by the fact that the satellite-based cloud retrieval is representative of the whole pixel, while the information if the sun is obscured over the ground-based station is representative for the (point) station, and it cannot be inferred from the satellite cloud retrievals. This combined with the facts that the direct irradiance attenuation from clouds is completely different from GHI, it is not linearly decreasing with cloudiness or cloud optical thickness and finally its contribution to GHI depends on various parameters (mainly solar elevation), introduces an issue in any instantaneous comparison between satellite based GHI retrieval representing the whole pixel and single point measured GHI. So, the main result of this sun visibility over the station analysis is to discuss on possible systematic biases due to satellite pixel versus station evaluation representativeness issue. This issue makes the instantaneous model output evaluation difficult, especially in partly cloudy situations.

Since the main source of errors in this analysis is associated with clouds, we further assess the satellite derived cloud input in the model. The MSG COT is transformed to CMFmsg using Eq. (2) and this is the cloud related input in the SENSE2 model. Since it cannot directly be evaluated with ground-based measurements, we indirectly evaluated the CMFmsg with the CMF derived from GHI measurements (Eq. 7) and the results are presented in Fig. 9, as relative frequency distributions of CMFmsg, CMF and their difference (CMFmsg – CMF), for all cases and different cloudiness conditions. Overall, the CMFmsg is overestimated (0.17, Fig. 9a and b), which is the reason for the overall overestimation of SENSE2 modelled GHI. This CMFmsg overestimation comes mainly from situations that are characterized as cloudy (Fig. 9g, h,i and j) and at the same time when the sun is obscured over the ground based station (Fig. 9m and n).

There are also cases of CMFmsg underestimation (differences of CMF <0 in Fig. 9b), which come mostly from situations characterized as cloudless (CMF≥0.9, Fig. 9e and f) and explains the points below the 1:1 line in Fig. 4a and 8, indicating that the measured GHI is greater than the modelled. The first reason for this is a cloudy satellite pixel, namely CFMmsg<1 in Fig. 9e, but the ground-based CMF=1 indicates that no clouds are over the station. There is also a large fraction of CMF > 1 (Fig. 9e) that is attributed to the irradiance enhancement by clouds, that occurs often when the sky above the ground station is partially cloudy, and the sun is visible (see also Fig. 9k and l and Fig. 8e). In this case, the reflection of solar radiation by clouds increases the diffuse component from directions relatively close to the sun, and hence the ground based measured GHI. This is a 3-dimensional effect of clouds that cannot be reproduced using 1 dimensional radiative transfer modelling that used in this study. This is a limitation of the SENSE2 model that does not include 3-dimensional cloud effects (enhancement of GHI or parallax) which can be reproduced using 3D RT simulations (e.g. Mayer 2009). However, the 3-dimesional cloud structure information is not available for an operational solar energy nowcasting model from geostationary satellites (Qu et al.,

2017; Schroedter-Homscheidt et al., 2022) and on the other hand the introduction of parameterizations and techniques to improve the computational time (Tijhuis et al., 2023) is essential.

Summarizing, to explain the overestimation of SENSE2 GHI retrievals, we have to consider that the direct comparison between point measurements of solar radiation at the ground and of satellite estimates representative of a pixel, introduces deviations (e.g., Kazadzis et al., 2009; Schenziger et al., AMT; Carpentieri et al., 2023) linked with the cloud features within this pixel and the limitations of cloud monitoring using satellite data (e.g. spatial resolution). We investigated both CMFs' distribution and their differences separately again for clear sky conditions according to the satellite (namely COT=0, Fig.9 c and d).

Regardless the unique value of CMFmsg=1 meaning no clouds resolved by the satellite, there are cloudy cases for the ground-based station with CMF<1 (Fig.9 c). Due to the satellite spatial resolution, at some cases, small scale broken clouds cannot be resolved (e.g., Schenziger et al., 2023, Marie-Joseph et al., 2013; Qu et al., 2017), but those clouds may have a significant impact in ground based measured irradiance in case that they are obscuring the sun (almost total attenuation of direct irradiance). In case that they do not obscure the sun is the clear sky case also for the ground-based station, without excluding

the effect of cloud enhancement of measured GHI (CMF=1 and CMF>1 in Fig.9 c, respectively). In a recent study by Schenziger et al. (AMT) using sky camera images, the limitation of MSG satellite based modelled CMF is demonstrated for small scale clouds. For two different stations inside the same satellite pixel that was characterized as cloud free different results were found. For one station that was cloud free the model agreed with measurements, while for the station that was covered by localized cumulus clouds that couldn't resolved by the satellite, lead to discrepancies between ground based and satellite

based modelled values. Nevertheless, even for the cases that the satellite imager can resolve clouds within a partially cloudy pixel, the COT product for this pixel is a constant value, namely a spatially homogeneous cloud optical property for the corresponding area. In this atmospheric scene with high spatial variability of clouds, GHI measured at the ground level, with the sun obscured or not, will affect the comparison dramatically.

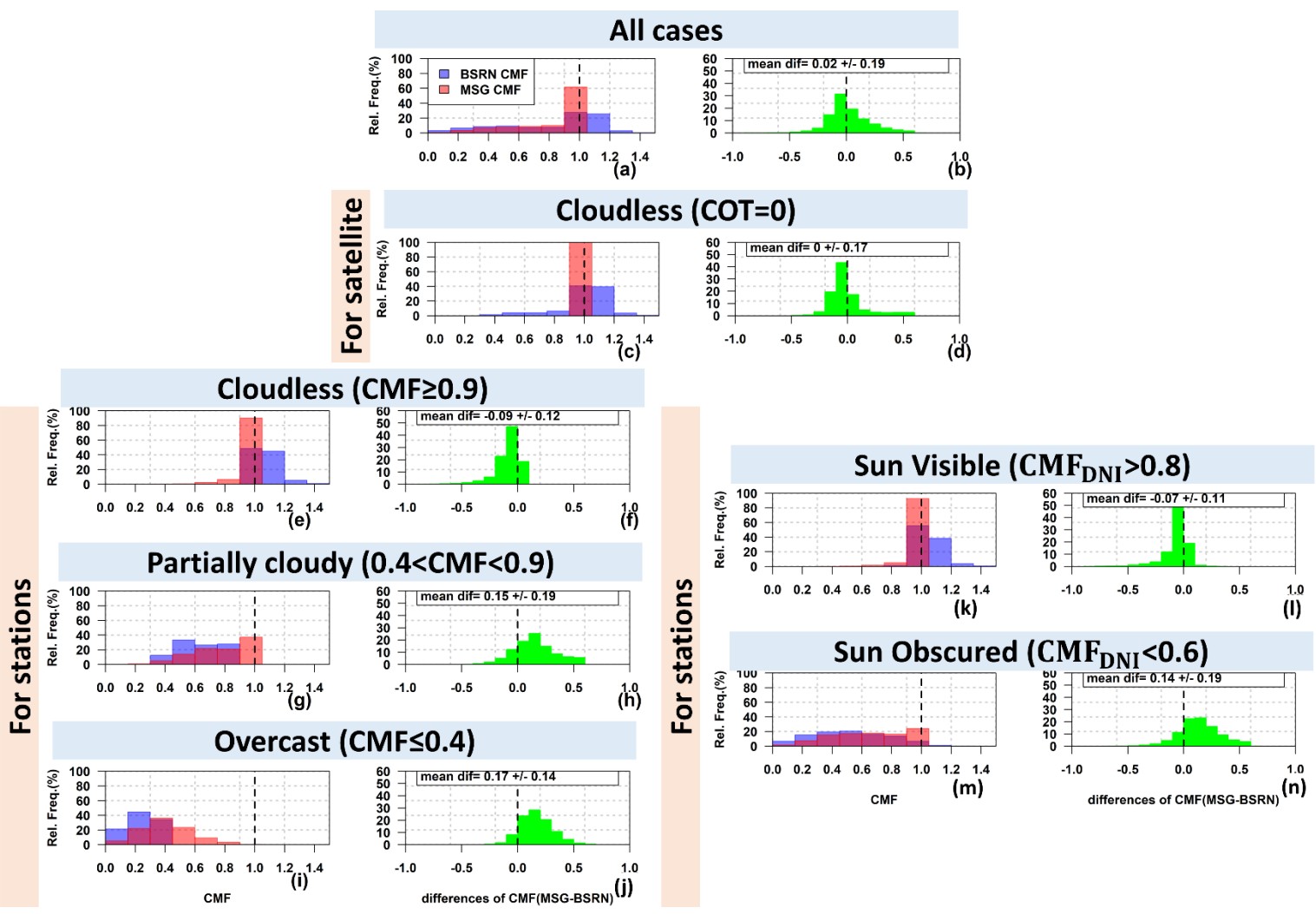

**Figure 9** Left panels (a, c, e, g, i, k, m): Distribution of cloud modification factor (CMF) from measurements of global horizontal irradiance (GHI) (blue bars) and from MSG cloud optical thickness (COT) (red bars), under all cases and under different cloudiness conditions. Right panels (b, d, f, h, j, l, n): Distribution of the CMF differences between those derived from measurements of GHI and those derived from MSG satellite COT.

### 3.1.4 Bias correction based on cloud input

The model overall relatively overestimates GHI, which is attributed to the CMFmsg overestimation. Based on the main conclusions from CMF differences from the previous section, we tried to find if there is any common pattern for all stations, in those CMF differences (modelled against measured) as a function of CMFmsg, since the latest is the only operationally available input, every 15min. Additionally, we found that those differences hardly change with SZA (Fig. B1 in the Appendix B), so we will investigate their relation only with CMFmsg.

We calculated the mean CMF difference and their standard deviation per CMFmsg bins for every station and the results are presented in Fig.10. A pattern of mean CMF differences was found for almost all station (apart from TAM, ATH and THE) with CFMmsg overestimation up to almost 0.1 starting from CMFmsg bin 0.3 up to 0.8, related also with low standard deviations over those bins.

   As we discussed in the previous section, this CFMmsg overestimation (up to ~0.1) is mostly related to partial cloudiness and
the sun obscured conditions over the station. Nevertheless, the sun's visibility is information that couldn't be provided by satellites. Consequently, we tried to correct CMFmsg (the operational input) with those CMF differences (modelled against measured). We used the mean of the of CMF differences per CMFmsg bin from seven out of ten stations (excluding TAM, ATH and THE) to derive the correction factor (the correction hereafter), which is depicted as the thick black dashed line in Fig. 10. The correction was applied to CMFmsg values falling in the bins 0.3-0.8 only. The correction was applied to all
stations, including TAM, ATH and THE, which act as a test bed (low frequency of cloudy cases) for the general correction derived from the rest seven stations.

   Table 3 summarizes the statistics of corrected modeled GHI against ground-based measurements. The MBE and RMSE are improved after the correction. LER and CAM are the two stations with the greatest improvement in their statistics, following CAB and LIN, which was anticipated since those are stations at higher latitudes associated with high cloudiness. Even ATH
and THE stations, that were independent from the correction factor derivation, exhibit better results after applying the correction. TAM is the only station where statistics weren't improved. Due to its rare cloudiness, this station's statistics were already good, indicating that probably a hybrid approach of the correction according to the area's cloudiness would be better. Overall, after the correction for 61% of the cases the GHI differences (modeled against measured) were within +/-50 $W/m^2$ (or +/-10%). The MBE for all stations was also improved to 11.3 $W/m^2$ (2.3%), compared to the uncorrected values (23.8 $W/m^2$
or 4.9%). For the daily mean GHI the overall MBE and RMSE was improved to 3.3 $W/m^2$ (1.7%) and 13.1 $W/m^2$ (6.6%), compared to the uncorrected values of 6.6 $W/m^2$ (3.3%) and 15.4 $W/m^2$ (7.7%), respectively. For monthly means, the MBE improved to 2.7 $W/m^2$ (1.6%) compared to 5.7 $W/m^2$ (3.2%) before correction and the RMSE to 6.3 $W/m^2$ or 3.6% (before correction 9.2 $W/m^2$ or 5.2%).

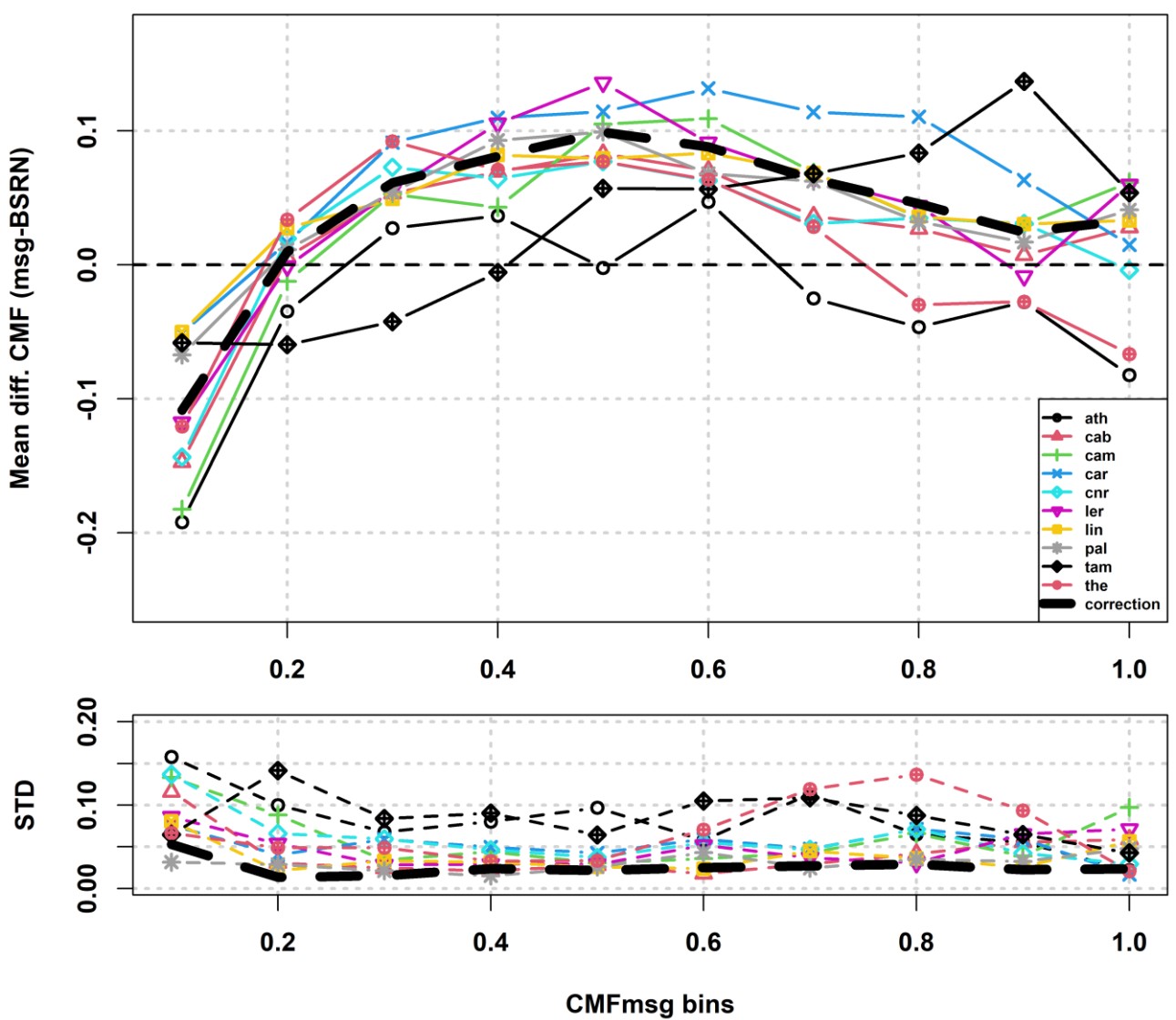

**Figure 10** Upper panel: Mean cloud modification factor (CMF) difference between modeled (from MSG satellite cloud optical thickness) values CMFmsg and those derived from global horizontal irradiance (GHI) measuremnts per modeled CMFmsg bins. Lower panel: The coresponding standard deviation (STD) of CMF differences per modeled CMF bins. The different colors in lines and different symbols correspond to different stations.

**Table 3 Performance of nowcasted irradiances before and after correction with CMFmsg.**

| | | 15min | | | | | | daily | | | | | | monthly | | | | | |
| | | MBE W/m² (%) | | RMSE W/m² (%) | | R | | MBE W/m² (%) | | RMSE W/m² (%) | | R | | MBE W/m² (%) | | RMSE W/m² (%) | | R | |
| station | N | | cor. | | cor. | | cor. | | cor. | | cor. | | cor. | | cor. | | cor. | | cor. |
|---|---|---|---|---|---|---|---|---|---|---|---|---|---|---|---|---|---|---|---|
| ATH | 9472 | 14 (2.5) | 8 (1.4) | 97 (17.5) | 97 (17.5) | 0.94 | 0.94 | 4.1 (1.9) | 2.1 (0.9) | 13.2 (5.9) | 12.3 (5.5) | 0.99 | 0.99 | 4.5 (2.3) | 1.8 (0.9) | 5.6 (2.8) | 3.3 (1.7) | ~1 | ~1 |
| CAB | 7749 | 31 (7.8) | 11 (2.9) | 115 (29.0) | 110 (28.0) | 0.88 | 0.88 | 9.2 (5.5) | 3.3 (2.0) | 15.2 (9.2) | 11.4 (6.9) | 0.99 | 0.99 | 8.0 (6.0) | 3.2 (2.4) | 9.3 (7.0) | 4.2 (3.2) | ~1 | ~1 |
| CAM | 2376 | 34 (10.1) | 13 (3.7) | 104 (30.8) | 98 (29.0) | 0.89 | 0.89 | 7.9 (5.8) | 3.6 (3.2) | 14.7 (10.8) | 11.5 (10.0) | 0.98 | 0.99 | 8.1 (6.6) | 3.7 (3.4) | 8.2 (6.7) | 4.5 (4.1) | ~1 | ~1 |
| CAR | 8985 | 30 (5.8) | 23 (4.5) | 84 (16.2) | 80 (15.4) | 0.95 | 0.96 | 9.5 (4.5) | 8.0 (3.8) | 14.5 (7.0) | 12.5 (6.0) | 0.99 | 0.99 | 7.9 (4.2) | 6.6 (3.5) | 9.9 (5.3) | 8.5 (4.5) | ~1 | ~1 |
| CNR | 8806 | 19 (3.7) | 6 (1.2) | 104 (20.6) | 102 (20.1) | 0.93 | 0.93 | 5.0 (2.5) | 2.1 (1.1) | 14.0 (6.9) | 11.1 (5.7) | 0.99 | 0.99 | 4.2 (2.3) | 1.8 (1.0) | 7.6 (4.2) | 4.7 (2.7) | ~1 | ~1 |
| LER | 5191 | 44 (12.9) | 21 (6.1) | 131 (38.6) | 124 (36.5) | 0.82 | 0.83 | 9.9 (7.0) | 3.6 (2.6) | 21.6 (15.3) | 17.9 (12.7) | 0.97 | 0.97 | 10.9 (7.6) | 4.1 (2.9) | 12.9 (9.0) | 5.4 (3.8) | ~1 | ~1 |
| LIN | 7989 | 39 (10.0) | 21 (5.3) | 119 (30.8) | 114 (29.6) | 0.88 | 0.88 | 12.4 (7.3) | 7.1 (4.2) | 18.4 (10.8) | 13.7 (8.1) | 0.99 | 0.99 | 10.3 (7.6) | 6.0 (4.5) | 12.2 (9.0) | 7.5 (5.5) | ~1 | ~1 |
| PAL | 8011 | 28 (6.7) | 11 (2.5) | 117 (27.4) | 113 (26.5) | 0.90 | 0.91 | 6.6 (3.5) | 1.9 (1.0) | 14.1 (7.4) | 11.6 (6.1) | 0.99 | 0.99 | 5.1 (3.4) | 1.4 (1.0) | 7.4 (5.0) | 4.9 (3.3) | ~1 | ~1 |
| TAM | 9011 | -8 (-1.2) | -12 (-2.0) | 98 (15.4) | 98 (15.4) | 0.94 | 0.94 | -3.9 (-1.5) | -4.1 (-1.6) | 18.5 (7.2) | 18.2 (7.2) | 0.94 | 0.94 | -3.7 (-1.5) | -4.5 (-1.8) | 9.8 (3.8) | 8.8 (3.5) | 0.97 | 0.97 |
| THE | 9188 | 27 (5.2) | 18 (3.6) | 91 (17.6) | 88 (16.9) | 0.95 | 0.95 | 8.2 (3.8) | 6.0 (2.8) | 13.6 (6.3) | 11.2 (5.2) | 0.99 | 0.99 | 6.7 (3.5) | 4.8 (2.5) | 9.1 (4.7) | 7.2 (3.7) | ~1 | ~1 |

After the improvements in the configuration of the SENSE2 model and by correcting the bias in CMFmsg for partially cloudy conditions (for CMFmsg bins between 0.3 to 0.8, the "bell-shaped curved" that has been also reported in other studies e.g. Marie Joseph et al. (2013)), more accurate estimates of GHI have been produced, in line with the results from similar models (Qu et al., 2014; Thomas et al., 2016; Qu et al., 2017). These SENSE2 GHI estimates will be the basis for the new forecasting system NextSENSE2, evaluated in the next Section (3.2).

Comparing out results with other studies, for HC3v3 database of surface solar irradiation (Qu et al., 2014), for the 15 min time scale, correlation coefficient values greater than 0.92 and relative RMSE between 14-38% were found. In the same study, for daily irradiation correlation coefficient values greater than 0.97 were found with relative RMSE between 6 and 20 %. For the latest version 5 of HelioClim-3 database (HC3v5) the validation against 14 BSRN stations (Thomas et al., 2016) resulted bias between -4 to 5% and rRMSE between 14.1 and 37.2% for GHI. Both studies highlight the good performance of the clear sky irradiation values from the McClear clear-sky model (which uses advanced inputs for aerosol, water vapor and ozone, instead of climatological values). The comparison of 15 min means of global irradiance estimated by the fully physical Heliosat-4 method (combination of McClear and McCloud models) against ground-based measurements from 13 stations of the BSRN network (Qu et al. 2017) showed large correlation coefficients for all stations (0.91-0.97), and bias and RMSE of GHI that ranged between 2-32 $W/m^2$ and 74-94 $W/m^2$ respectively. In the same study, the greatest values of relative RMSE of the mean irradiances were found for stations with rainy climates and mild winters (26% to 43%, the greatest value for the northernmost station), while for stations in desert and Mediterranean climates values ranged between 15% and 20%, which are in line with our finding for the northernmost and southernmost stations in this study. The previously observed positive biases using APOLLO cloud retrieval in Heliosat-4 (Qu et al. 2017) for CAMS Radiation Service, have been significantly reduced and balanced after applying the new cloud retrieval scheme of APOLLO_NG (new cloud mask with cloud probability threshold to 1% among other improvements, more details in Schroedter-Homscheidt et al., 2022). After the improvements, a relative RMSE of hourly GHI between 10.3 and 25.5% with a mean of 13.7% has been reported for 2015 (Schroedter-Homscheidt et al., 2022). An extensive validation (Urraca et al., 2017) of the operational product (ICDR) of the CM SAF over Europe for 2008-2015 period gave for daily means of the product a MBE 4.5 $W/m^2$ (4%) and RMSE 18.1 $W/m^2$ (15.1%) and it was reported that it was overestimated at high latitudes in contrast to the climate data records (CDR). For the new SARAH-3 CDR SIS product for the period 1983-2020 the validation (Pfeifroth et al., 2023) showed for the 30-min instantaneous data, daily mean, and monthly mean biases of 4.2 $W/m^2$, 2.18 $W/m^2$ and 2.25 $W/m^2$ respectively. The validation of the operational product (ICDR) with respect to the SARAH-3 CDR for the year 2020 showed that it consistently extent the SARAH-3 CDR in time. The reasons for differences between these two products were the different auxiliary data (like water vapor, etc.) and time range used for driving effective cloud albedo and daily snow cover.

## 3.2 Short term forecasting

### 3.2.1 Overall performance - Benchmark with Persistence method

Figure 11 summarizes the performance of the CMV method (green points) in predicted GHI as function of forecasting horizon, by providing main statistics, after comparison with GHI ground based measurements, from all ten stations, for a whole year (2017). Detailed results per station, for representative statistics and selected time steps (+60, +120, +180 min) can be found in Table 4. As a benchmark, the results of the commonly used persistence forecasting method are presented also in Fig. 11 (black points). We can see that the CMV model systematically outperforms persistence for all time steps. It is interesting that the first

time step (+15 min) is not the one with the maximum difference between the CMV and persistence statistics (or the maximum of CMV FS%), indicating that for such short time interval the probability of changing cloudiness is low, which favours the persistence method. The second time step is the one with the maximum of CMV FS% (best performance) compared to persistence up to ~10%. As the forecasting horizon increases all metrics deteriorate for both methods, while, at the same time, persistence is systematically worse than CMV.

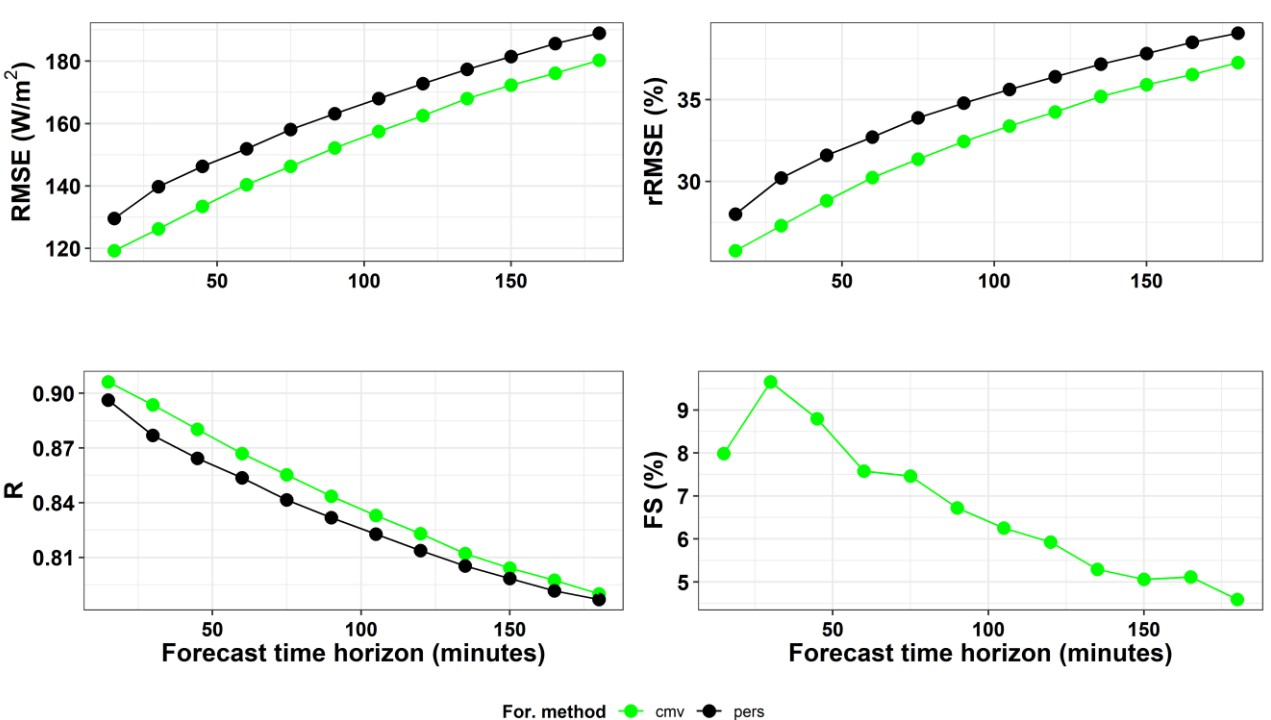

**Figure 11 Performance statistics of CMV modeled (green points) and persistance method (black points) forecasted global horizontal irradince (GHI) for every 15 min time step up to 3 h ahead.**

**Table 4 Performance statistics of CMV modeled forecasted global horizontal irradince (GHI) for the 60 min, 120 min and 180 min time steps.**

| station | Mean CMF | rRSME (%) time step (min) | | | R time step (min) | | | FS (%) time step (min) | | |
|---|---|---|---|---|---|---|---|---|---|---|
| | | +60 | +120 | +180 | +60 | +120 | +180 | +60 | +120 | +180 |
| ATH | 0.97 | 22.0 | 24.5 | 25.3 | 0.90 | 0.87 | 0.86 | 0 | 1.7 | 3.2 |
| CAB | 0.68 | 39.7 | 45.6 | 49.6 | 0.80 | 0.73 | 0.68 | 10.5 | 7.5 | 5.8 |
| CAM | 0.63 | 54.1 | 62.0 | 68.9 | 0.66 | 0.58 | 0.50 | 2.7 | 3.4 | 1.5 |
| CAR | 0.85 | 22.8 | 26.3 | 29.6 | 0.90 | 0.86 | 0.82 | 8.1 | 5.6 | 3.1 |
| CNR | 0.81 | 31.0 | 34.8 | 37.4 | 0.85 | 0.79 | 0.75 | 2.4 | 2.9 | 2.6 |
| LER | 0.61 | 50.8 | 56.0 | 59.6 | 0.73 | 0.67 | 0.62 | 10.4 | 9.5 | 9.4 |
| LIN | 0.68 | 39.7 | 45.9 | 51.2 | 0.81 | 0.74 | 0.69 | 13.9 | 10.9 | 8.8 |
| PAL | 0.68 | 38.9 | 44.9 | 48.8 | 0.82 | 0.75 | 0.71 | 11.6 | 7.6 | 6.0 |
| TAM | 0.87 | 21.4 | 23.1 | 24.7 | 0.89 | 0.86 | 0.82 | 2.2 | 1.0 | -1.3 |
| THE | 0.91 | 23.9 | 27.2 | 29.2 | 0.89 | 0.86 | 0.84 | 6.8 | 5.1 | 2.4 |

An interesting grouping of stations resulted by comparing main statistics (rRMSE and FS) for both forecasting methods with stations' mean CMF, representing their mean cloudiness (Fig.12). Three time-steps were selected (+60, +120, +180 min) and are depicted with increasing transparency. Two groups of stations are evident. Those with high mean cloudiness (LER, CAM, PAL, LIN, and CAB), which show worse rRMSE than those of lower cloudiness (ATH, THE, TAM, CAR and CNR), independently of the method. Again, the CMV model (green symbols) outperforms the persistence method (black symbols) for all stations for these time steps (except of TAM for +240 min). The interesting finding is that the FS (%) of the CMV method increases with decreasing CMF, namely the forecasting skill of CMV model is higher compared to persistence for stations with higher cloudiness, demonstrating the applicability of the CMV forecasting method on GHI under cloudy conditions.

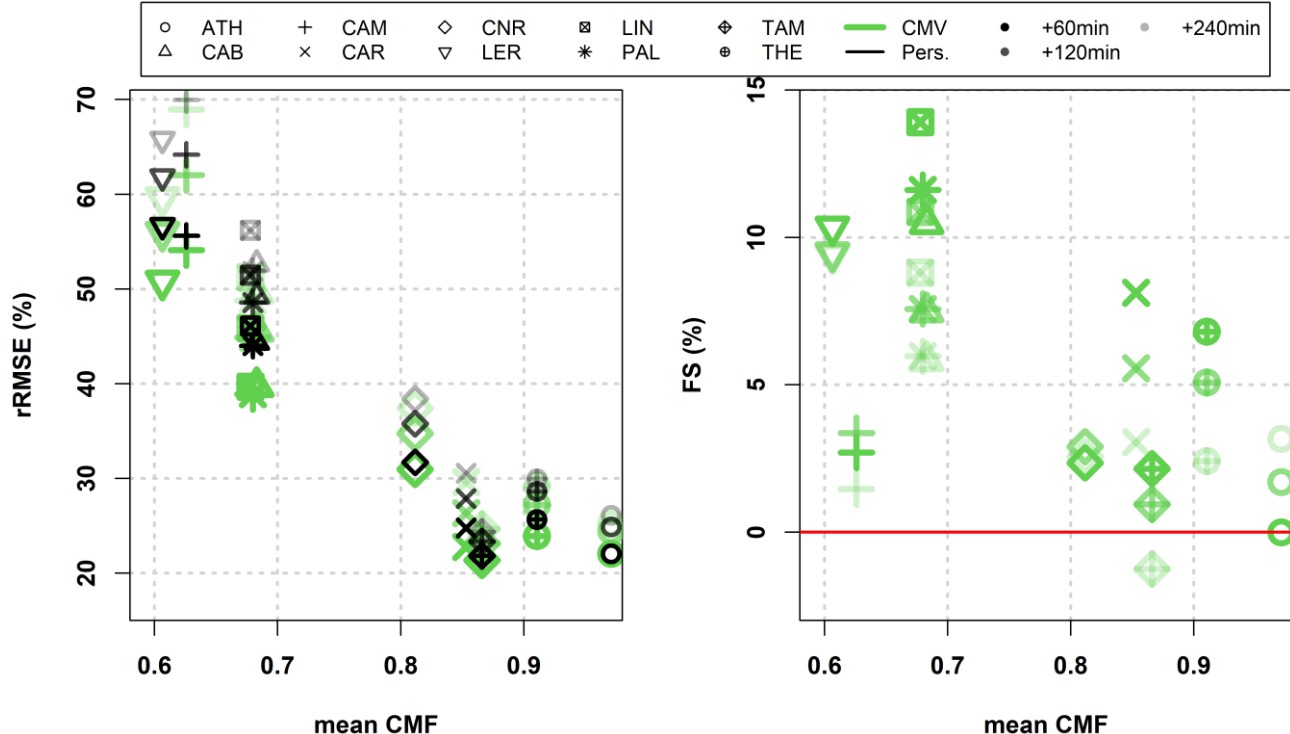

**Figure 12 Mean bias error (MBE) and its relative values (rMBE%), relative root mean square error (rRMSE%) and forecasting skill expressed in percentage (FS%) of CMV model (green symbols) and persistence method (black symbols) forecasted global horizontal irradiance (GHI) versus the average cloudiness of stations (mean CMF) for the time steps +60, +120 and +240 min (increasing transparency of symbols).**

### 3.2.2 Performance for different cloudy conditions

To demonstrate the value of the CMV model against the persistence method that assumes the same cloudy conditions for all future time steps, we compare their performance under different cloudy conditions and transitions in cloudiness. Figure 13 presents the RMSE for both models (CMV green points and persistence black points) and the CMV model FS% as a function of CMF, for three time-steps (+60, +120 and +180 min). Persistence performs better than CMV model under clear sky conditions, namely CMF=1, for all times steps (as expected, as there is no change in cloudiness). This is also true for the CMF

bin 0.9 only for time step +180 min and for CMF bin >1 for all time steps, a bin which contains mainly clear sky cases. For cloudy conditions, namely CMF<0.9, the CMV model outperforms persistence for all time steps (apart from +180 min time step and CMF bin 0.9). The cloudier the conditions (the smaller the CMF), the better the performance of CMV model and the

greater the CMV FS% (up to ~ 20% for time step +60 min). The FS of CMV model decreases slightly with forecasting horizon, however, for the maximum of the forecasting horizon (+180 min) remains quite high ~+10% for CMF bins <0.7.

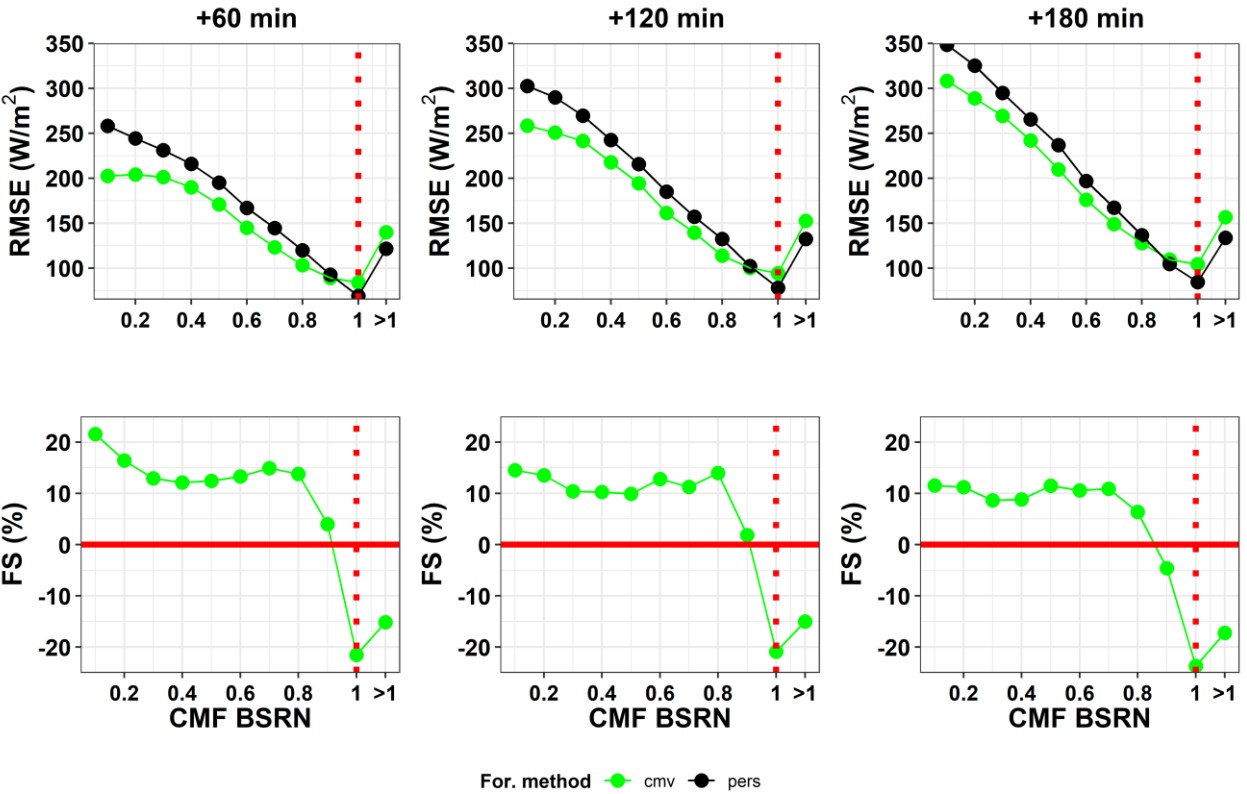

**Figure 13 Upper panel: Root mean square error (RMSE) of forecasted global horizontal irradiances (GHI) for all stations with the CMV model (green symbols) and with the persistence method (black symbols) versus cloud modification factor (CMF) derived from GHI measurements, for 3 time steps (+60, +120 and +180 min). Lower panel: The CMV model forecasting skill against CMF classes for the same time steps with upper panel.**

To demonstrate the better performance of CMV method compared to the persistence for all time steps under cloudy conditions, we calculated CMV FS% for partially cloudy conditions (0.4 < CMF<0.9) and overcast conditions (CMF≤0.4) and the results are presented in Fig. 14. We can see again that the FS of CMV model decreases with time, however, the minimum value is ~10% for both categories. The maximum of FS is for both categories at +15 min time step at ~16% and ~22% for 0.4<CMF<0.9 (cross symbols) and CMF≤0.4 (triangle symbols), respectively.

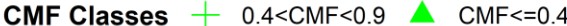

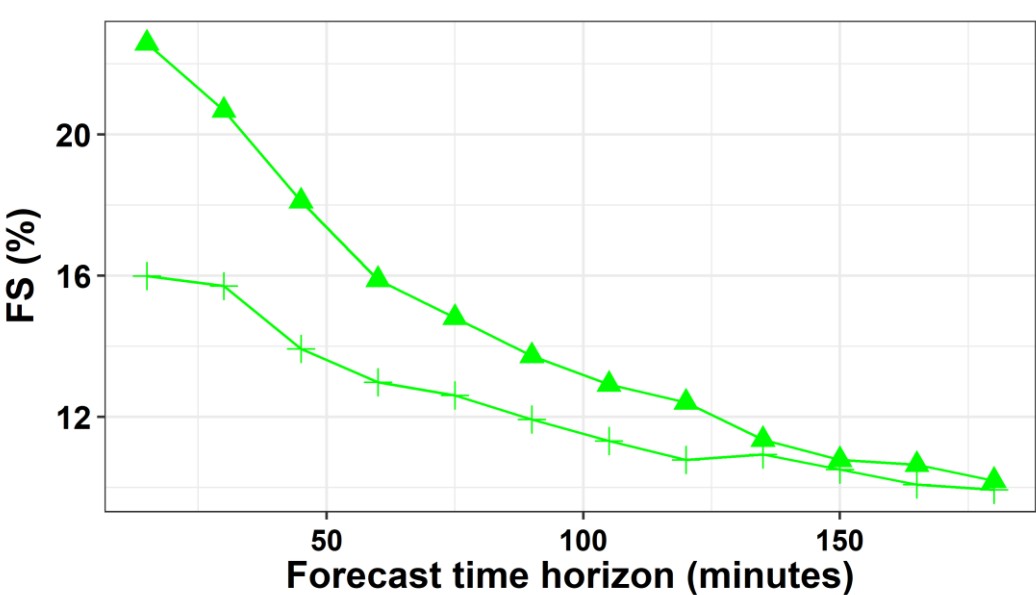

**Figure 14 Forecasting skill (FS) expressed in percenateges of CMV model against the persistence method as a function of time horizon, from all stations, for two different cloudiness conditions: 0.4<CMF <0.9 (crosses) and CMF ≤0.4 (triangle).**

The performance of the CMV model against the persistence method was assessed also for changing cloudiness, in terms of CMF changes (from ground-based measurements). The CMF changes were calculated for a time interval of 60 min (as $\Delta CMF=CMF_{t+60}$ - $CMF_t$), and the results of the CMV model FS (%) are presented for the +60 min time step, as a function of CMF changes in Fig. 15. The high negative value of FS for zero CMF changes bin indicates that the persistence method is better for that bin, which was anticipated, since we have zero or almost zero changes of CMF, which practically is the persistence method definition. Persistence is still better than CMV for CMF changes from cloudy to clearer conditions, up to the +0.3 CMF change bin, but the FS is less negative than the zero bin. For CMF changes with higher magnitudes (bins > 0.4) from cloudy to clearer conditions, CMV is better that persistence, with FS values 15% for the CMF change bin +0.6. Consistent results were found for the opposite situation, namely from clearer to cloudy conditions, with CMV model always being better that persistence, with FS values up to ~20%.

Our analysis for different cloudiness conditions, highlights the limited ability of the persistence method compared to the CMV based NextSENSE2 to accurately forecast GHI under cloudy conditions (CMF values<0.9) and to follow the transitions in cloudiness (especially from clearer to cloudy conditions).

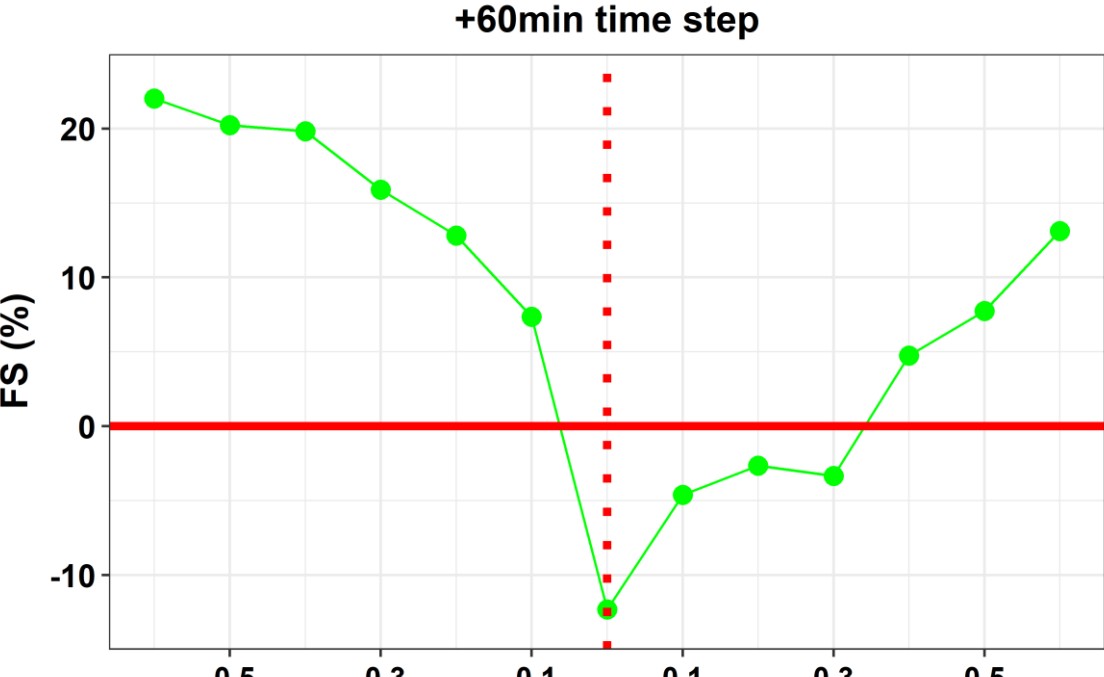

**Figure 15 Forecasting skill (FS) expressed in percenateges of CMV model against the persistence method from all stations as a function of CMF changes within 60 min time interval (from time 0 to +60 min time step).**

A direct comparison of the results of the present study for the NextSENSE2 short-term forecasting model with other studies is not straightforward, as the study period, the geographical area and the validation methods are different. Kallio-Myers et al. (2020) validated their Solis-Heliosat satellite-based GHI forecast modelled over southern Finland and they found that the rRMSE reached 50% at 4h time step. Urbich et al. (2019) validated SESORA short-term forecast of solar surface irradiance over Germany and parts of Europe for seventeen different cases with different weather patterns for the period August to October 2017, with all forecasts initiated at 9:15 UTC and reported for the validation against the SARAH-2 data by CM SAF a RMSE of 59 W/m$^2$ after 15 min which reaches its maximum (142 W/m$^2$) after 165 min.

One of the limitations and source of error for the NextSENSE2 is related to the satellite based optical flow method for the short-term forecasting and it is that it cannot reproduce cloud formation or dissipation. One example is the convective clouds that form very fast, violating the criterion of the optical flow of constant intensity of the pixels between two consecutive images (e.g., Urbich et al., 2018; 2019). Urbich et al. (2018) applied the common approach of the separation into subscales for the optimization process, which eventually didn't improve the forecast, while increased at the same time the complexity of the implementation. As has already been discussed in the introduction, satellite based short term forecasting is the best choice for the time horizon up to 6h ahead, since it is available in real time and at high spatial resolution. However, its merge with NWP models it is a solution for increasing the time horizon and the quality of forecasts (Lorenz et al., 2012; Wolf et al., 2016)

compensating for the effect of changes in intensities (during convection or cloud dissipation) that cannot be captured by CMV models (Müller and Pfeifroth, 2022). A comparison has been presented by Urbich et al. (2019) between a short-term forecasting model of surface solar radiation (the SESORA model) with different NWP models (apart with the persistence model) and found that the intersection point that the NWP model delivers better results depends on the model and it is beyond 3-4h, which is also in line with the finding of other studies (e.g., Lorenz et al., 2012; Wolf et al., 2016). The merge of our short-term forecasting model with NWP model is out of the scope of the present study. After the elaborative benchmark analysis of the NextSENSE2 system with persistence approach, its applicability as an operational tool for the time horizon up to 3h ahead has been demonstrated.

## 4 Summary and conclusions

Our motivation is the continuous improvement of the EO based estimates and the accuracy of short-term forecasts of available solar resources to support solar energy exploitation systems on a regional scale (Europe and MENA region). In this study, we improved the SENSE/NextSENSE nowcasting/short-term forecasting operational systems, and analyzed in detail the cloud related uncertainties, discriminating also situations based on sun visibility, using ground-based measurements.

In terms of the aerosol related inputs, the slight overestimation of CAMS AOD that was found against the AERONET retrievals (<10%) resulted to SENSE2 clear-sky GHI underestimation lower than 1%, highlighting the applicability of CAMS forecasts as EO inputs for operational solar resources nowcasting. In terms of modeled all skies GHI, it was found that SENSE2 mostly overestimates GHI with MBE 23.8 W/m$^2$ (4.9%) for instantaneous comparisons, which was attributed to the uncertainties related to satellite cloud retrievals (overestimation of CMFmsg by ~0.17) and also to the spatial representativeness between satellite based retrievals and ground-based measurements. We demonstrated that the most difficult situations to be modeled are related to high spatial variability of solar radiation within the satellite pixel due to clouds (e.g., small broken clouds and the sun obscurity over the ground station, an information not possible to be derived from satellite data). Based on our cloud related analysis using ground-based data, a correction for the modelled GHI was used, resulting to an overall improvement of the SENSE2 modelled GHI with 61% of the cases within +/-50 W/m$^2$ (+/-10%) of measured GHI and a final MBE of SENSE2 11.3 W/m$^2$ (2.3%). Our main analysis was based on the 15 min time scale, however based on the application hourly, daily or monthly data could be used. The daily and monthly SENSE2 GHI showed much better statistics (MBE 3.3 W/m$^2$ and 2.7 W/m$^2$, respectively). The validation results of SENSE2 demonstrate high accurate nowcasted values of GHI which are in line with similar models. The recorded positive bias could be reduced by applying improvements in the NWC SAF cloud retrieval input to the SENSE2 regarding partially cloudy pixels. NextSENSE2 was also improved due to the SENSE2 improvements. We also show that compared to the persistence method, the model works much better (as expected) at locations with increased cloudiness and for frequent cloudiness changes.

The data and methods involved for the estimation/prediction of the GHI in this study also reveal their limitations. As mentioned, the pixel-based approach of the model inputs (satellite and models) could not always reflect the reality above a

(point) ground-based station. However, the model inputs are the state of the art of EO data and can be readily available in regional or global scale, at high spatial and temporal resolution, hence the GHI product is representative of an area (~5km x 5km in this model), which is useful for PV parks covering a wider area. In general, evaluating the performance of such EO based GHI models with ground-based measurements must account for these comparison spatial representativity issues. The optical flow algorithm for calculating CMVs is also based on assumptions like 2D clouds and brightness constancy. However, it is a method based on cloud inputs from satellite data in real time and the applicability of those methods is demonstrated here compared with the persistence approach.

Since satellite cloud information is the only real time input, a new straightforward configuration for estimating GHI was applied (SENSE2). The advantage of calculating clear sky GHI from the previous day, is what increases the accuracy of this product, since it is based on a detailed LUT of ~16M combinations of 7 different inputs, considering apart from AOD, additional aerosol optical properties and atmosphere/surface state inputs. Thus, the uncertainties in the estimated clear sky GHI practically result only from uncertainties in the model inputs. The new scheme for calculating all skies GHI by multiplying the clear sky GHI with CMFmsg (derived in real time by multiparametric function of MSG COT and SZA) was improved by applying a suitable CMFmsg correction. The correction was successful and improved the model performance, especially for areas with high cloudiness. Additionally, the new configuration of the SENSE2 is more flexible, and it is easy to adapt and provide more products like DNI, UV index, PAR, etc. which is one of the prospects for the new model to run in a retrospective way, using reanalysis data or in situ observational data for certain locations.

According to the results, high resolution (every 15 min, at ~5km x 5km) and quite accurate GHI real time estimates/forecasts are produced from the upgraded SENSE2/NextSENSE2 operational systems that can contribute to solar energy systems management and planning.

**Appendix A**

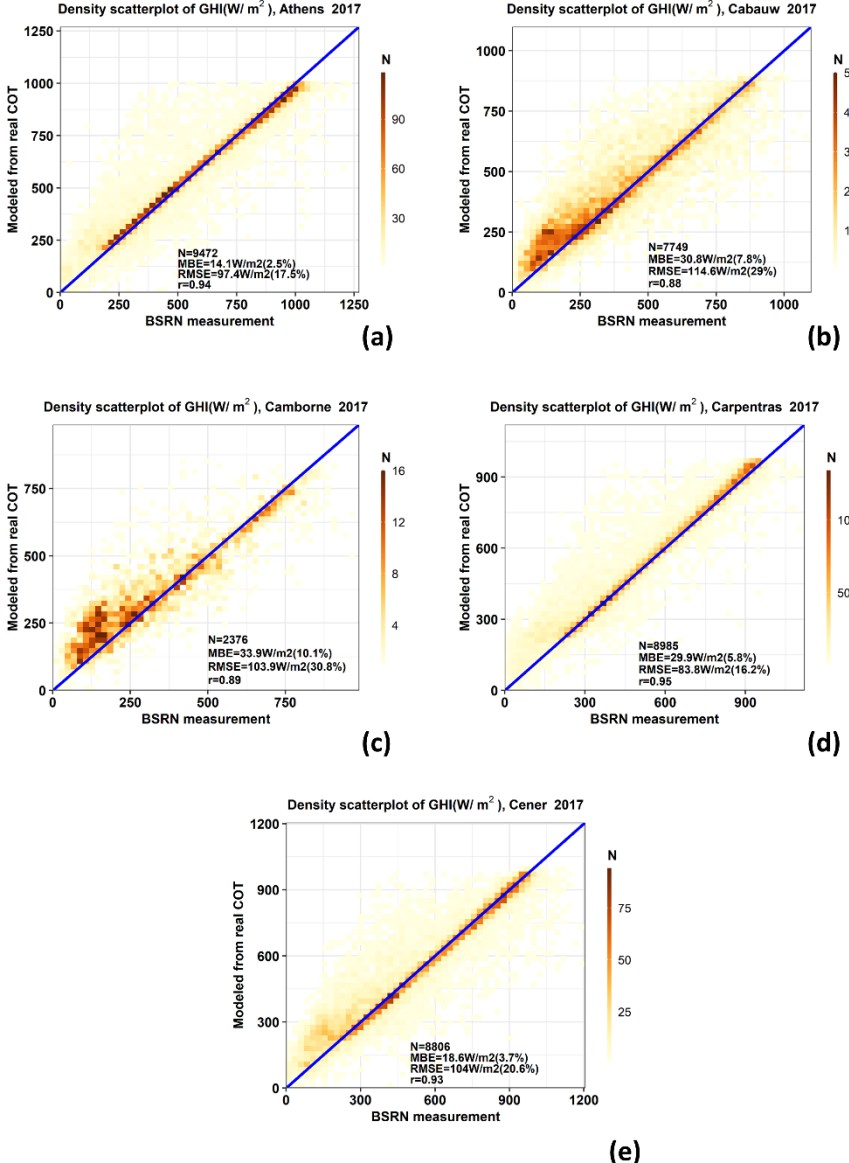

**Figure A 1 Comparison of the global horizontal irradiance (GHI) modeled versus measured for (a) Athens, (b) Cabauw, (c) Camborne, (d) Carpentras and (e) Cener, for 2017.**

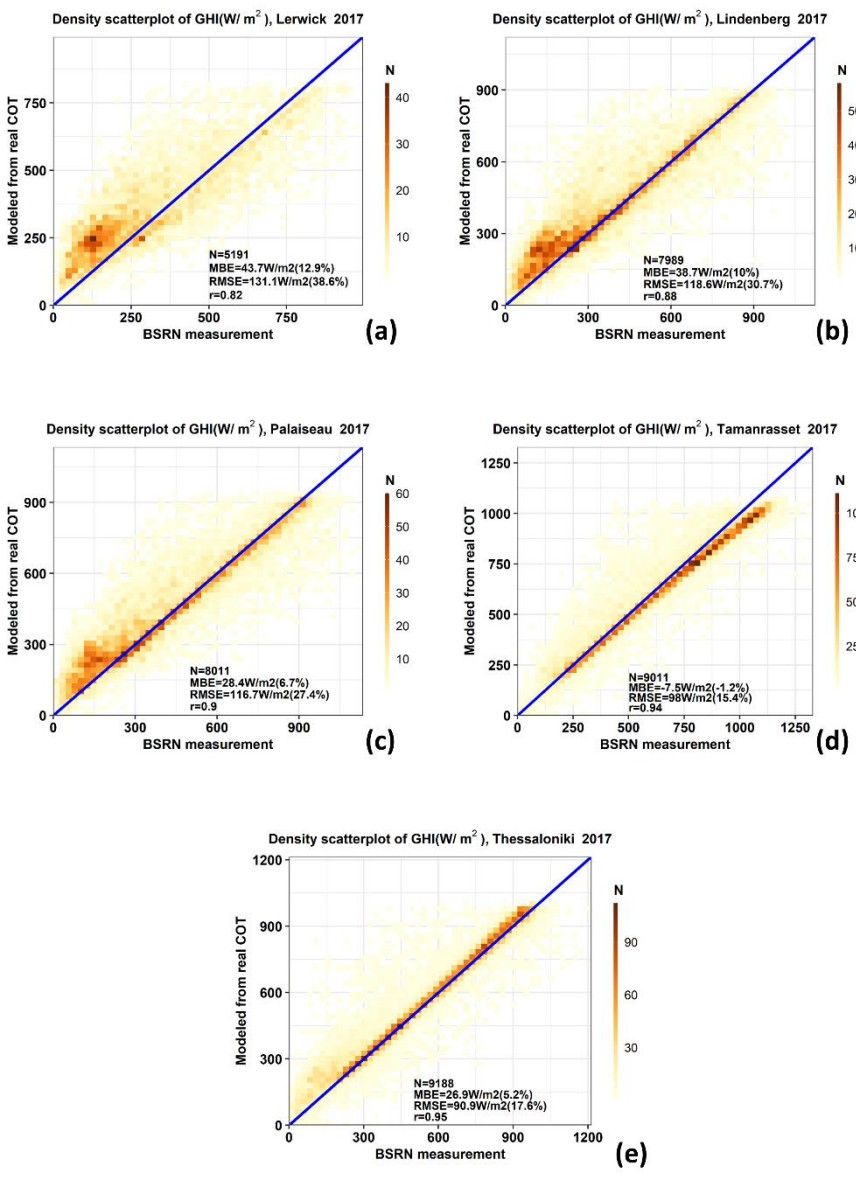

**Figure A 2 Comparison of the global horizontal irradiance (GHI) modeled versus measured for (a) Lerwick, (b) Lindenberg, (c) Palaiseau, (d) Tamanrasset and (e) Thessaloniki, for 2017.**

**Appendix B**

To see if the CMF differences (MSG modelled against measured) changing with SZA, the MBE of CMF was calculated for bins of SZA every 10 degrees. The observed CMF was considered the one derived from GHI measurements (Eq. 7) and the modelled one derived by the Eq. 2. The results are presented in Fig. A1, for all cases and under different cloudiness conditions,

along with the relative values of CMF MBE expressed in percentages. We can see again the fact that the main overestimation of CMF values by MSG COT comes from the cloudy conditions (CMF <0.9). Specifically, for the partially cloudy conditions (0.4<CMF <0.9) the MBE reach values up to ~ 0.20 and for overcast skies (CMF ≤0.4) there are SZA bins (0 and 70 degrees) that the MBE reach values up to 0.4. However, for those two categories the MBE hardly changes with SZA.

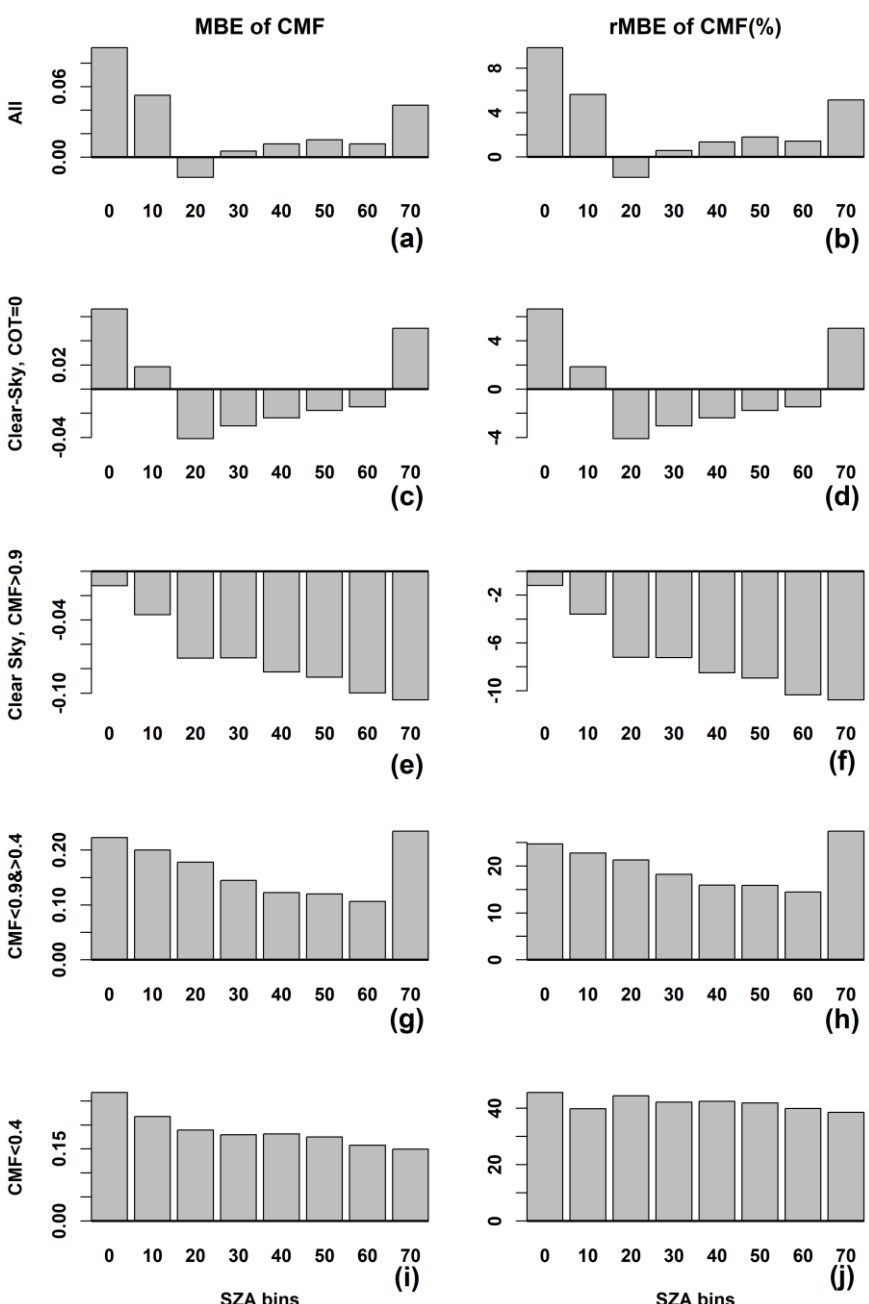

**Figure B 3 Cloud modification factor (CMF) mean bias error (MBE – left column) and relative MBE (% - right column) as a function of solar zenith angle (SZA) under all cases and under different cloudiness conditions.**

**Author contributions.**

Idea and initialization, KP and SK; Model parameterization, IF, KP, IPR; resources, AFB, BEP, IPR, CK; data provision and curation, AFB and BEP; cloud function approach, NP and AK; overview and revision, CK, MH, SK; 1st draft writing, visualization, analysis and interpretation, KP; writing, review and editing, all authors. All authors gave final approval for publication.

**Acknowledgments.**

We would like to thank the 8 site instrument operators and technical staff of the BSRN network stations who made the ground-based measurements feasible. We acknowledge the principal investigators and co-investigators and their staff for establishing and maintaining the 10 AERONET sites used in this study. This study contains modified Copernicus Atmosphere Monitoring Service information [2023], and neither the European Commission nor ECMWF is responsible for any use that may be made of the Copernicus information or data it contains. We acknowledge the Eumetsat SAFNWC services as well as the OMI team for providing all the necessary data used in this study. We acknowledge the free use of the GOME-2 surface LER database provided through the AC SAF of EUMETSAT. The GOME-2 surface LER database was created by the Royal Netherlands Meteorological Institute (KNMI). This work was supported by computational time granted from the National Infrastructures for Research and Technology S.A. (GRNET) in the National HPC facility - ARIS - under project ID pa210301- SO-LISIS. CK, SK, IF and KP would like to acknowledge the European Commission project EuroGEO e-shape (GA no. 820852). AK would like to acknowledge the co-financing by the European Union and Greek national funds through the Operational Program Competitiveness, Entrepreneurship and Innovation, under the call RESEARCH – CREATE – INNOVATE (project code: T1EDK - 00681). SK, IF and KP would like to acknowledge the COST Action HARMONIA, CA21119, supported by COST (European Cooperation in Science and Technology).

**Financial support.**

This research has been supported by the European Commission project 'EXCELSIOR': ERATOSTHENES: Excellence Research Centre for Earth Surveillance and Space-Based Monitoring of the Environment (grant no. 857510) and the European Commission project EuroGEO e-shape (GA no. 820852).

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
