# Peer review of "Effects of clouds and aerosols on downwelling surface solar irradiance nowcasting and sort-term forecasting"

_Atmospheric Measurement Techniques, 2023_

## Author Comment (AC1)

**Reply to anonymous Referee #1**

We would like to thank anonymous referee#1 for his/her comments, that helped us improve the manuscript. In the following, analytical replies are provided to each of the reviewer's comments. Reviewer's comments are written in bold font. Line numbers, when provided, refer to the new version with track changes.

**The study reflects the lack of adequate reflection on previous work and literature in this area. However, this should be the basis and first step of developments and publications. This lack leads to misleading interpretations of the results and introduction of concepts as novel, which are in contrast well known. I cannot recommend this manuscript for publication in the current form. Please see the detailed comments for further information. I can not recommend the manuscript for publication, see comments below. However, the evaluation results might motivate the publication of the manuscript after a proper major revision.**

Thank you for your time and comments, they certainly helped towards a better-quality manuscript.

**Major comments/concerns:**

**L105 and others:**

**The author uses "~16M combinations of simulated GHI at the earth's surface". However, several concepts exist with smarter solutions, e.g. with the hybrid eigenvector concept the amount of needed RTM calculations can be reduced to several hundred, also the good old NREL etc. approximations works well, see https://doi.org/10.1016/j.rse.2009.01.012 or https://doi.org/10.3390/rs4030622 and references therein for further details. A respective discussion or proof of the benefits of the approach presented in the manuscript is missing. Thus, it is not clear why this algorithm is needed and what the benefits compared to other well established methods are.**

**Reply**

*We acknowledge that different concepts exist to reduce the amount of RTM simulations that the LUTs needed to describe the surface solar radiation for different atmospheric/surface conditions and hence reduce the computational time without significantly compromise the accuracy of the retrieval. However, the main concept behind the generation of our clear sky LUT was to have, in addition to total*

*shortwave, also spectral irradiance outputs in order to be able to integrate different spectral regions according to the targeted application (e.g. erythema, vitamin D, photosynthetically active radiation).*

*Technically, since the operational set up of the SENSE2 model allows for the computation of the clear sky GHI values from the previous day, the processing time for interpolation to the 7 dimensions of the LUT has no effect in producing timely the real time output of the model every 15 min, while the accuracy of the clear sky output is almost identical with direct RTM simulations, which was investigated in a previous study focusing on clear sky conditions (Papachristopoulou et al., 2022). So, in this case the uncertainties of the clear sky GHI retrievals are related mainly to the uncertainties of the model inputs.*

*In addition, the 16M simulations were performed only once for building this LUT. This LUT includes various aspects especially for aerosols (aerosol optical depth, single scattering albedo, Ångström Exponent) that can reduce the uncertainty under different aerosol conditions and for broadband solar radiation or specific spectral regions. Especially in the shorter wavelength regions under cloud free conditions and in the presence of different aerosol types.*

*Finally, we follow similar LUT approach with one that has already published in the previous study by Kosmopoulos et al. (2018), using less combinations. The relevant discussion was added in the manuscript (lines 229 – 247 in the new version of the manuscript with track changes attached) presented previous studies, new concepts along with the proof of the benefits of the approach adopted by the current study.*

**L 125 "....Wdata (Bhartia, 2012) based climatology) and surface albedo (GOME-2 database (Tilstra et al., 2017, 2021))**

**Several important aspects are not discussed which affects the accuracy of the product. Is the used SAL consistent with that used for COD ? Further, is the aerosol information used for the COD retrieval identical to those used for SIS ? Are the BRDF corrections for SAL and COT identical ? If not, what does that mean for the consistency of the product. Please note, in particular inconsistent SAL data can lead to a significant bias in SIS, this could be the reason for your bias and not the "sun obscuration".**

Reply

*In the revised version of the manuscript more details were including regarding those products, specifically in Lines 221-222 that the directionally dependent*

*Lambertian-equivalent reflectivity product of GOME-2 was used and in Line 258-259 the relevant references for the cloud retrievals.*

*Regarding the retrieval algorithm of cloud optical properties for the 2017 dataset used in this study (Météo-France, 2016):*

- *The surface characteristics are obtained from monthly climatology over land or constant value over sea. Specifically, over land surface reflectances were derived from MODIS white-sky surface albedo monthly climatology available from NASA ([http://modis-atmos.gsfc.nasa.gov/ALBEDO/index.html](http://modis-atmos.gsfc.nasa.gov/ALBEDO/index.html)). These white sky albedos represent bi-hemispheric reflectances without the direct component which is a good approximation of the surface albedo below a cloud. Over sea, constant values were used: 3% (at 0.6mm) and 1% (at 1.6mm).*
- *An aerosol free atmosphere is assumed, which might introduce errors under high aerosol plumes (like desert storms and biomass burning events) if a single channel approach used for the retrieval of cloud information (Mueller et al., 2015), but in our case those errors are eliminated (~60% of dust cloud cases are detected) because the retrieval algorithm is based on a multichannel approach and differences in channels according to the cloud features and respective spectral signatures (Météo-France, 2021).*
- *We used for the GHI retrievals directionally dependent Lambertian-equivalent reflectivity (DLER) of the Earth's surface retrieved from GOME-2 satellite observations (Tilstra et al., 2021). The surface DLER describes Lambertian (isotropic) surface reflection which is extended with a dependence on the satellite viewing geometry. This is a global database of surface reflectivity for 26 wavelength bands between 328 and 772 nm as a function of the satellite viewing angle via a second-degree polynomial parameterization. According to Tilstra et al., (2021) although DLER and BRDF surface reflectances have different properties, they are comparable for wavelengths $\lambda > 500$ nm. Based on their results, the GOME- 2 surface DLER is compared with MODIS surface BRDF data from MODIS band 1 (centered around 645 nm) using both case studies and global comparisons and they concluded that the GOME-2 DLER compares well to MODIS BRDF data. So, the surface albedo used in this study is consistent with the dataset used for the retrieval of cloud optical properties.*

*We also performed a sensitivity for the effect of surface albedo to GHI and for cloudless conditions 10% variation in surface albedo led to <1% variation in GHI, and under cloudy condition (following Fig. 1) it led less than 3 and 5% variation in GHI for COT=2 and 12 respectively.*

[Figure]

Figure 1. Percentage difference (%) in global horizontal irradiance as a function of surface albedo (using as reference zero value for surface albedo) and solar zenith angle (SZA) under cloudy conditions (cloud optical depth – (a) COT =2 and (b) COT =12) and an atmosphere including aerosols (aerosol optical depth – AOD =0.5 and single scattering albedo- SSA=0.95.)

Additionally, according to our results the error in satellite derived CMF has no correlation pattern with used surface albedo for all stations.

**L137, Eq.1**

**The so called Cloud Modification Factor is the good old clear sky index or cloud coverage index (Cano et al), used in several EU projects and SAFs long ago, ranging from SODA, Satellight to Heliosat-3 and CM SAF. Respective references should be given. Further, by introducing this factor to correct bias resulting from COD they proof that the direct path (see https://doi.org/10.5194/amt-15-1537-2022 and references therein) is more favorable. It is not clear why the authors have chosen the indirect path.**

**Reply**

*The Heliosat method and the semi-empirical models using the combination of the clear sky index (derived by cloud coverage index from satellite data) and a clear sky model is a well-established method, and it is used by all the projects mentioned by the reviewer and according to the suggestion all the respective references have been included in the manuscript by adding in the introduction 1 paragraph (lines 52-56 from the old version of the manuscript were replaced by lines 75 -98 in the new version of the manuscript).*

*To justify the choice of the fully physical approach and the use of cloud optical properties retrievals instead of the direct path, 2 additional paragraphs were included in the introduction (lines 99 -125) and the paragraph 3 of the previous version of the manuscript (Lines 65-74) has been changed accordingly (Lines 126-146 in the new version of the manuscript).*

*The main rationale for this choice is that the original SENSE was designed as a fully physical model taking advantage of the NWCSAF cloud product generated operationally in house using MSG data and up to then, as far as we know, hasn't used before from other solar energy nowcasting application. We decided to retain this fully physical approach at SENSE2 and at the same time improve the scheme that uses that specific cloud input to retrieve surface solar irradiance.*

**L189:**

**Optical flow method *"We apply Farnebäck".* Is Farneback still state of the art for SSI nowcasting ? In Urbich et al 2018 (https://doi.org/10.3390/rs10060955) evidence is given that TV-L1 outperforms Farnebäck. This finding is supported by other studies and the maths. TV-L1 is more robust concerning changes of the intensities. It is part of OpenCV and thus free software as well. Further, it is not enough to compare your nowcasting method to persistence. It should be compared with other methods as well, including state of the art NWP (see respective publications of the IEA framework). Within this scope the effect of changing intensities on the quality of the forecasts should be discussed. One of the first works in the area of solar surface irradiance (SSI) nowcasting can be traced back to Lorenz et al. (University of Oldenburg, now ISE) and others. These works should be cited and discussed as well.**

Reply

*The current study is a follow-up study of Kosmopoulos et al. (2020). In this study both methods Farnebäck and TV-L1 used to forecast clouds (in terms of COT and CMF) and compared against persistence approach (clouds at the same position) for 3 (pan-European) case study days with different cloud movement patterns. Both methods performed equally well, with Farnebäck showing slightly better results for the selected test days, highlighting the strong dependence of the accuracy of both methods to the specific characteristics of the cloud patterns. It was pointed out that the selection of Farnebäck after the optimization of the model parameters it wasn't a generalization, and further analysis was needed in terms of comparing irradiances (forecasted against ground-based measurements) for at least one full year of forecasts for more robust conclusions. For more information for the reader*

*Lines 320-323 have been added in the new version of the manuscript. We would like to follow up this comment and during a future study to evaluate both optical flow methods in a larger spatiotemporal timeframe using SENSE2. However, it is a difficult task to isolate optical flow uncertainties, as such analysis has to isolate first other uncertainties based on cloud properties retrievals and assumptions and also to take into account regional to local aspects linked with e.g. cloud formation and flow above various terrains and prevailing atmospheric conditions (e.g. different cloud types and levels).*

*The discussion regarding the comparison of our short-term forecasting method with other methods beyond persistence, including state of the art NWP and the effect of changing intensities on the quality of the forecasts has been added at section 3.2 Lines 742-764 of the new version of the manuscript.*

*According to the suggestion of the reviewer the works in the area of solar surface irradiance (SSI) short-term forecasting have been cited and discussed. We introduce a new paragraph in the introduction (Lines 56-63 of the previous version of the manuscript were replaced by Lines 146-165 in the new version of the manuscript).*

**L 315** ***"The interesting part is that the same case stands for the whole range of measured GHI, indicating that it is a general limitation of satellite that it cannot take into account clouds."*** **as well as** **L 334.** ***"Cases with partial cloudiness and the sun obscured as seen from the ground sensor (almost total attenuation of direct irradiance) will be associated with low measured irradiance that cannot be captured by the model. This is the main reason of the overall model overestimation."***

**Misleading discussion and interpretation. The satellite can of course take into account clouds that obscure the sun. Else, all values would be clear sky values, or ? Let us assume a partly cloudy pixel with 50 % clear sky and 50 % cloudy sky, leading to an average cloudiness of 0.5. If only the sun obscured regions (100% cloud sky) are investigated you surely will find a bias, namely, an overestimation of SSI by the SAT retrieval, because the area average seen by the satellite is partly cloudy (50%). But statistically, there are also situation where the ground based station sees the sun (100% clear sky), but the satellite is partly cloudy sky (50%). On average there is an "error" cancelation of these effects. No figure or statistics are shown for situations where ground measurements see the sun, but the pixels are partly cloudy. Hence, there is no proof that the overall bias results from the "sun obscured" effect. In several studies bias values are reported for algorithms without sun obscuration correction, which are not significant or depending on the method positive or negative (see e.g. validation**

reports and publications of CM SAF, e.g. Uccarra et al, https://doi.org/10.1016/j.rse.2017.07.013). Thus, the cancellation of the "errors" induced by different viewing geometries seems to work well and they are several other reasons for the bias. Thus, your conclusion seems a bit hasty and misleading. You should check SAL, it is likely a source for your bias.

In addition, there already exists a lot of publications dealing with broken clouds, 3-D cloud effects or the uncertainties arising from the comparison between ground based and satellite based SSI. Please read them, discuss and cite them and clarify what your work adds to existing woks. In my opinion currently not much, beside misleading conclusions. Of course, for slant geometries the cloudiness is overestimated, but that is another story, which is not taken into account in your study. You will find respective articles, e.g. in the CM SAF publication list.

Reply

*Probably it was not clearly written but of course using the satellite-based inputs here is the reason to take clouds into account, so of course satellite data can take into account clouds. It is just some limitations that are discussed here. The main problem comparing instantaneous measurements is exactly that under cloudy conditions satellite pixel and ground-based point measurements have differences mainly attributed to the sun visibility or not. The discussion on this point is clearer in the new text.*

*What is meant here is that satellite-based cloud retrievals cannot distinguish between cases that the sun is obscured and cases that it is not over the ground-based station. This combined with the facts that direct sun attenuation from clouds: a. is completely different from GHI, b. it is not linearly decreasing with cloudiness or cloud optical thickness and finally c. its contribution to GHI depends on various parameters (mainly solar elevation), introduces an issue in any instantaneous comparison between satellite based and measured GHI. So, it is mainly a spatial representativeness issue that affects the model evaluation statistics.*

*About the example and the statement of the 50% of the pixel covered lead to a an "error cancelation". Yes, it does spatially lead to a possible error cancelation, but it does not statistically when we use a single point measurement to evaluate the instantaneous (15 min) satellite pixel-based retrieval. Exactly due to the nonlinear behavior of direct sun irradiance to cloud coverage. In a very simplistic way for a 50% cloudiness with a calculation of a COT =X: The average GHI of the cases that the clouds obscure the sun and the ones that is not obscured, is not equal with the case of half cloudiness or using 50% of X (it is less due to the Direct ~ exp(μCOT) dependence (μ is the air mass)). So yes, in this case there is a 50% chance statistically that the sun is obscured and 50% that is not. But errors in GHIs for all*

*these cases are not cancelled out. Things are more complex with cloudiness of more or less than 50% and also depends on the solar elevation ranges (which in addition are location dependent e.g. higher latitudes - less range). However, as said the purpose of satellite-based use models is to represent the whole pixel that for a lot of cases is not represented by a single point station. So, the main meaning of this sun obscure or not analysis is to discuss on possible systematic biases due to this representativeness issue.*

*Here is an example of 2 and a half years of 1-minute pyranometer (CMF) and camera images (cloud fraction) and pyrheliometer (sun-visibility) measurements at Davos, Switzerland.*

[Figure]

*Figure 2 Cloud modification factor (CFM) based on 1-minute pyranometer data as a function of cloud fraction (CF) derived from sky camera image, for sun visible and sun obscured conditions over the ground-based station (based on pyrheliometer data).*

*Blue and orange are "two different worlds" to estimate using satellite data, and for e.g. 50% CF, probably having a number of pyranometers in a satellite pixel would capture the mean pixel irradiance as retrieved based on the satellite cloudiness. However, for instantaneous comparisons, a 50% cloudiness as seen from the satellite will lead to a reflectance or COT retrieval that will not lead to the calculation of a mean (of the blue and orange curve) CMF. So, satellite-based CMF will never be either orange (given that COT>0) or blue. In our opinion it will be systematically lower than the blue and orange average CMF as in the case of visible sun a 50% cloud covered pixel COT will underestimate more GHI (through direct sun large underestimation) than the obscured sun case overestimation. It is still a representativeness comparison issue and not a satellite "problem", but maybe it is interesting to mention it here. The respective discussion has been added in Lines 524-531 of the new version of the manuscript to complete the discussion regarding the sun visibility analysis over the ground-based stations.*

*However, we agree that the whole section needs clearer writing, and we thank the reviewer for the comments that help on this direction. The statement "satellite cannot take into account clouds that obscure the sun" may lead to misleading impression since it was incomplete. It just meant that for a partial cloudiness the*

*satellite does not "know" in which one of the above Figure 2 curves, the pyranometer instant measurement belongs.*

*Both sentences that reviewer is mentioning in his comment have been changed accordingly, along with the whole Section 3.1.3 major changes in conjunction with the comments of referee 2 (the reason for GHI bias and the discussion of the results moved to the end of 3.1.3, at the CMFmsg analysis).*

*With the sentence of L315 we would like to say that due to the satellite spatial resolution, small broken clouds maybe cannot be resolved, resulting to COT=0 value, but those clouds may have a significant impact in ground based measured irradiance in case that they are obscuring the sun (almost total attenuation of direct irradiance) (new Fig. 8 d points above the identity line). In case that they do not obscure the sun is the clear sky case, but for GB measurements enhancement maybe occur. So, we replaced the sentence in L315 of the previous version of the manuscript with the following in Lines 497 -499 in the new version of the manuscript:*

"Most of the cases are on the 1:1 line, with few ones being higher, especially, for measured GHI<250 $W/m^2$, meaning that there are clouds over the ground-based station that haven't been resolved by the satellite pixel (COT=0)."

*We agree with referee's example of partially cloud pixels and the 1s case, and actually this is the main source of overestimation of our GHI retrieval, depicted in new Fig.8 e, f in combination with new Fig.8 b, c and new Fig. 9.*

*Of course, there are also cases with the opposite, partially cloudy pixels where the ground-based station sees the sun, which are depicted in new Fig. 8e for Sun visible situations as points below the 1:1 line.*

*But in our case, it is not evident the "error" cancellation (Fig. 4a and new Fig. 8) for the reasons that we discussed in the first part (paragraphs 1-5) of this answer regarding the instantaneous comparisons and due to COT product and how it treats the partially cloudy cases directly related to the satellite spatial resolution (the size of the clouds can be resolved).*

*To our opinion new Fig. 8e for Sun visible situations is the figure showing situations where ground measurements see the sun (as actual ground-based measurements have been used to define this) and the fact that the pixel is partially cloudy demonstrated by points below 1:1 line also possibly related with cloud enhancement. Meanwhile the Sun obscured situations at new Fig 8f are associated with high overestimation. At this point we would like to clarify that we didn't perform sun obscurity correction.*

*We included and discussed the proposed studies and the reported biases along with the reasons for the bias in the introduction (Lines 116 -125) and at the Section 3.1.4 where we compared our results of satellite GHI retrievals with other studies (Lines 634 -663).*

*The effect of SAL on the GHI bias has been investigated in the Major comment L125 of reviewer 1.*

*The relevant discussion and respective publications dealing with the uncertainties of the comparison between point ground-based measurements and satellite retrievals (Lines 570 -573), broken clouds (Lines 573 -585) and 3d effects (Lines 560 -569) have been added in the revised version of the manuscript.*

*In addition, old Fig. 8 now Fig. 9 has been updated to include the distributions of CMFmsg and measured CMF and their differences also for the classification of the cases for the sun obscurity to support the discussion in section 3.1.3.*

**The aerosol study is well done, but also this part lacks a bit on discussion and citations concerning former works**

Reply

*According to referees comment the respective discussion was added at 3.1.2 at Lines 456 -472 of the new version of the manuscript, as follows:*

"An overestimation of the CAMS forecasted AOD at 550nm is also reported for 2017 over Europe (average modified normalized mean bias ranging from ~10 to 30%) from the continuous quarterly evaluation of the AOD forecasts against daily AERONET cloud-screened (i.e. Version 3 level 1.5) sun photometer data (Basart et al., 2023; Eskes et al., 2021). While this is the case on average, in contrast during high aerosol loads, CAMS forecasted AOD is underestimated, especially in desert regions and during dust events (Basart et al., 2023; Papachristopoulou et al., 2022) which might explain the almost zero bias for Tamanrasset station (the overestimation of small AODs masked out by the frequent underestimation of large AODs) compared to the greater values of bias (>0.01) found for most of the rest stations. Qu et al. (2017) analysed case studies at Tamanrasset and found that the CAMS (MACC) AOD at 550nm is frequently underestimated against AERONET data during summer dust events, explaining the strong positive bias they found for their modelled direct irradiance (using Heliosat-4 method and the McClear clear sky model). In contrast to the CAMS AOD underestimation during dust events, in the same study (Qu et al., 2017) a systematic overestimation of AOD was found during periods free of those events for the two examined desert stations (Sede Boqer and Tamanrasset), to which they associated the underestimation of their modelled direct irradiance. The updated McClear v3 clear sky model used in study by Schroedter-Homscheidt et al. (2022) and for their GHI estimates under clear-sky conditions a negative bias was found for most of the station especially

for those located in dust affected regions, which is in line to our results although not directly comparable since they compared directly with the BRSN measured irradiances. Our results demonstrate the good performance of the clear sky model using CAMS forecasts, highlighting that AOD product forecasted by CAMS is suitable for GHI nowcasting applications.*"*

**Minor comments:**

**L54 ,"..considered as big data". Please delete, it is not really big data compared to other fields....**

Reply

*It has been deleted in the context of major revisions of the introduction for the major comment 3 of the reviewer.*

**L 47 "The availability of solar resources is primarily affected by clouds and aerosols (e.g., Fountoulakis et al., 2021; Papachristopoulou et al., 2022)."**

**This is misleading. In areas with low aerosol variability water vapor is much more important than AOD (as a climatology value works well there). Please add H20 as important variable.**

Reply

*Water vapor was also added as an important variable, by changing the previous sentence, now Lines 51-54, as follows:*

"Under all-skies the availability of solar resources is primarily affected by clouds (e.g., Fountoulakis et al. 2021) and for clear-sky conditions it depends on the atmospheric composition with the most important variables being aerosols (e.g., Papachristopoulou et al., 2022) and water vapor (Yu et al., 2021)."

**L95 "SENSE2 is an operational system that produces fast estimates of GHI in real time every 15min, for a wide area including Europe and Middle East-North Africa (MENA)"**

**Please mention how the user can get these data.**

Reply

The following sentence has been added in Lines 194-195:

"The new version of the SENSE2 system is available as a webservice via https://solar.beyond-eocenter.eu/#solar_short (last access: 2023-12-15)."

**L113: The aerosol model of Shettle is used, but no discussion of the limitation induced by the assumption of spherical aerosols is given.**

Reply

*Aerosol assumptions in any real time forecasting model with no real time measurement inputs are uncertainty sources. There are different "levels" of uncertainties concerning aerosols as inputs on solar forecasting models. Trying to keep this discussion short:*

*The profile: Shettle assumes a nearly exponential decrease in the extinction coefficient with altitude, which is not always realistic, but does not practically affect the GHI at the surface (e.g., Fountoulakis et al., 2022).*

*Shape and size: According to Song et al., 2022, Kok et al., 2017 and Fountoulakis et al. 2023 under review) this can be important above mineral dust related areas and in this case the size plays a much more important role than the shape. In the latest and references therein, in the case of spheroid dust particles the effect was found negligible, while for very large particles differences at the surface for GHI shortwave were found less than 1% over ocean and less than 5% over desert for solar zenith angles 0 to 30 degrees.*

*AOD: All the above can be clearly masked by the moderately high uncertainty of the AOD forecast in case of variable aerosol cases/locations.*

*Other than AOD properties: Any model has to use some kind of assumption on the aerosol type at the particular time and location. Here, as reported, we directly consider optical properties (SSA, Angstrom exponent) from a monthly climatology dataset. There are uncertainties related to the use of climatological optical properties which are minor relative to the uncertainty in the forecasted AOD, for the usual AOD levels over Europe. For higher AOD values (e.g., AOD (at 500 nm) > 0.3 - 0.4) uncertainties in aerosol optical properties (SSA) play a more significant role. Angstrom exponent can play a role on the GHI model calculation uncertainty in cases of locations with variable aerosol types and sizes, as most GHI estimation models do not use spectral AOD as an initial input.*

*In a few words in a GHI reanalysis study (using as model inputs aerosol measured optical properties) maybe aerosol size information, especially in dust aerosol areas,*

*could slightly improve the results. However, in a real time forecast model with AOD forecasts and other optical properties climatology, any aerosol shape correction is statistically meaningless.*

**L 140**: **Use of NWC SAF products: I did not understand the sense of this approach, why do you need NWC-SAF ?**

Reply

*The use of the cloud product of NWC SAF is explained in the introduction in Lines 132-139 of the new version of the manuscript.*

"It is a combination of geophysical input parameters from satellite-based and model data sources and a neural network (NN) technique, trained on precalculated surface solar radiation simulations (look up table – LUT) using RTM. It uses the cloud optical thickness (COT) retrievals produced by the Application Facilities Support to Nowcasting and Very Short Range Forecasting (NWC-SAF) algorithm based on the MSG satellite data and aerosol optical depth (AOD) forecasts from the Copernicus Atmospheric Monitoring Service (CAMS) as inputs to the NN to derive the surface solar radiation in real time. More details about the previous version of the SENSE service can be found in Kosmopoulos et al. (2018)."

**L 145:** ***Typical values for the effective radius (Reff = 10 µm) and the liquid water path (LWP = 1 g/m3 145 ) were used, given the unavailability of those data and their small impact on GHI"***

**This phrase is quite misleading. First of all there are algorithms available to derive Reff and LWP, further I would not say that the impact is small, in particular when considering ice clouds.**

Reply

*We thank the referee for his comment, it is true that the sentence gives a wrong impression. A more analytical description is given in the new version of the manuscript (Line 264-274), clarifying that the unavailability of the cloud information referred to the exact position and extent of the clouds inside the atmosphere. Also, the respective discussion and references are given regarding the impact of different cloud parameters on simulating surface solar radiation.*

"The design of the cloud model was a trade -off between the relevance of the cloud property and the operational implementation of the model. It has been shown in previous studies (Qu et al. 2017) that for most of the cases (except for high surface

albedo values >0.9), the cloud vertical position and extent has a small or negligible influence for the RTM simulations of surface solar irradiance. Under cloudy conditions, COT is the variable that has the greatest impact on simulating surface solar radiation (Qu et al., 2017, Oumbe et al., 2014, Taylor et al., 2016). In our simulations, spherical droplets were assumed, with typical values for the effective radius (Reff = 10 μm) and typical climatological mean heights (base at 2 km, 3 km height) (Taylor et al. 2016, Kosmopoulos et al., 2018), given the unavailability of height descriptors in the operational mode and the negligible influence of changes in droplet effective radius with respect to COT on simulating surface solar radiation (Oumbe, 2009) and towards simplify the cloud model. The COT of the cloud layer is additionally specified at 550 nm, which leads to an adjustment of the liquid water content default value of 1 g/cm$^3$, using the parameterization by Hu and Stamnes (1993)."

**L 195*: "Smart persistence"*: I find the term irritating, please delete it. Please clarify that this kind of persistence is typically used for SIS nowcast comparisons. Add some discussion and references of former works here as well.**

Reply

*The title of Section 2.3 changed to Persistence forecast and a more detailed discussion has been included in Lines 329-333 of the new version of the manuscript as follows:*

"2.3 Persistence forecast

It is not easy to evaluate the quality of different forecasting methods of surface solar radiation using only statistical metrics, since the study period, the geographical area and other factors are affecting their forecasting accuracy. That's why it is a typical practice of evaluation to benchmark the different forecasts against some simple forecast methods (Pelland et al., 2013). We used the persistence forecast to benchmark the CMV forecasted GHI of NextSENSE2 system which is a commonly used reference in solar forecasting (e.g. Kosmopoulos et al., 2020; Kallio-Myers et al., 2020)."

[revised manuscript text omitted]

---

## Author Comment (AC2)

**Reply to anonymous Referee#2**

We acknowledge anonymous referee#2 for his/her very useful comments, that helped us improve the manuscript. In the following, analytical replies are provided to each of the reviewer's comments. Reviewer's comments are written in bold font. Line numbers, when provided refer to the version with track changes.

**The manuscript describes improvements to two high-spatial resolution models used for the prediction of the surface global horizontal irradiance (GHI) over the area of Europe and Middle East-North Africa. The two models, in particular, are:**

- **SENSE2, a nowcasting system based on look-up-tables (LUTs) calculated using libRadtran radiative transfer model that uses as input the cloud optical thickness (COT) obtained from Meteosat Second Generation (MSG) satellite and aerosol optical depth (AOD) predicted by the Copernicus Atmospheric Monitoring Service (CAMS);**
- **NextSENSE2 is a short-term forecast (up to 3 hours ahead) system using the GHI of SENSE2 and the CMV technique for forecasting the satellite-derived COT.**

**The two model performances are validated against ground-based measurements of GHI carried out in sites belonging to the Baseline Surface Radiation Network (BSRN) in the area covered by the models and two additional sites in Greece. Measurements refer to 2017.**

**The analysis is mainly aimed at investigating the role of aerosols and clouds, atmospheric factors with large spatial and temporal variability, on the estimated GHI.**

**The prediction of short- and very-short-term GHI is one of the fundamental issues related to the efficiency of renewable energy-based systems, and the study described in this manuscript is in principle useful in supporting the development and optimization of these systems.**

**I think the manuscript should undergo major revisions before publication, addressing the issues highlighted as major and minor comments.**

**Major comments**

**Any references to published paper describing similar GHI prediction models or investigating the role of clouds and/or aerosols on GHI nowcasting/forecasts are missing. Thus the reader in not able to understand the goodness of the performance of the models presented.**

**A description of similar models should be presented in the introduction, as well as in the "summary and conclusions" paragraph the results of this work should be compared with those of similar studies conducted in the same study area or in different regions.**

Reply

*The manuscript has been substantially changed in various places (in the new version of the manuscript with track changes attached) especially in the introduction based on this comment. Missing parts that the reviewer describes have been added to the manuscript. Specifically:*

*- In the introduction 3 additional paragraphs (lines 52-56 from the old version of the manuscript were replaced by lines 75 -125) have been added, describing satellite estimates of GHI including real time services.*

*- The comparison of our work with similar studies regarding GHI satellite estimates has been added in Section 3.1.4 (lines 634 -663) analytically and in the summary and conclusion section in a more condensed format (lines 782 -785).*

*-In the introduction 1 additional paragraph (lines 57-63 from the old version of the manuscript were replaced by lines 147 -165) has been added, describing GHI forecasting models.*

*-The comparison of our work with similar studies regarding GHI short-term forecasting based on satellite CMV models has been added at the end of Section 3.2 (lines 742 -748).*

**The performance of the nowcasts in paragraph 3.1.1 is not well supported. The sentence "This overestimation is attributed to the underestimation of cloud related information from satellite (MSG COT), when we compare point measurements with a pixel in satellite images corresponding to a wide area of almost 5 km x 5 km" needs to be argumented because no evidence of COT underestimation is supported here.**

**Moreover, the authors attribute the model's overestimation of BSRN measures for low GHI values to stations with more cloud cover,**

particularly those at high latitudes, such as Lerwick. However, evidence of the cloudiness in the various sites is not provided and the results are not presented for a single station. In my opinion a GHI scatterplot similar to that of Fig 4a for individual stations could be added as supplementary material.

Reply

*We agree with the reviewer that we are not talking about "underestimation of satellite COT" since there is not a direct analysis for COT, so this was corrected throughout the manuscript. We assessed CMFmsg against ground-based CMF to assess the error in SENSE2 GHI estimates due to uncertainties in the cloud input. We are also talking about a systematic statistical overestimation of GHI at certain cases, due to the different spatial representativity of satellite based and single point station irradiances and the non-similar behaviour of GHI and direct solar radiation when the sun is obscured or not.*

*We also agree with referee regarding the sentence that is not argued at the point given in the previous version of the manuscript, so we moved the discussion (the deleted Lines 402-407 and 408-411 of Section 3.1.1) of the reasons of the modeled GHI overestimation at Section 3.1.3 where the satellite cloud information of CMFmsg is evaluated against ground-based CMF (lines 544 -588).*

*To support the performance of the nowcasts in paragraph 3.1.1 an extra figure was added (new Figure 5 in the new version of the manuscript) relating the model bias and mean measured GHI with cloudiness and latitude for various sites. The discussion of this new figure was added in Lines 416-425. In addition, a GHI scatterplot like that of Fig 4a for individual stations is not provided in Appendix A.*

**The discussion of paragraph 3.1.2 on the aerosol effects on cloud-free GHI should be completed with the appropriate references addressing the CAMS and AERONET AOD comparisons.**

Reply

*According to referees comment the appropriate references addressing the CAMS and AERONET AOD comparison and the respective discussion was added at 3.1.2 at lines 456 -462 of the new version of the manuscript, as follows:*

"An overestimation of the CAMS forecasted AOD at 550nm is also reported for 2017 over Europe (average modified normalized mean bias ranging from ~10 to 30%) from the continuous quarterly evaluation of the AOD forecasts against daily AERONET cloud-screened (i.e. Version 3 level 1.5) sun photometer data (Basart et al., 2023; Eskes

et al., 2021). While this is the case on average, in contrast during high aerosol loads, CAMS forecasted AOD is underestimated, especially in desert regions and during dust events (Basart et al., 2023; Papachristopoulou et al., 2022) which might explain the almost zero bias for Tamanrasset station (the overestimation of small AODs masked out by the frequent underestimation of large AODs) compared to the greater values of bias (>0.01) found for most of the rest stations."

**Minor comments**

**Line 30:  use "significantly improved" instead of "improved a lot".**

Reply

*This changed in the new version of the manuscript (Line 30).*

**Line 51: add a sentence on the large temporal and spatial variability of clouds and aerosols.**

Reply

*A sentence was added in Lines 54-55 of the new version of the manuscript:*

"Among those variables, clouds and aerosols are characterized by large temporal and spatial variability which constitutes them as key variables for solar energy applications."

**Line 70: change "form" with "from".**

Reply

*Done.*

**Lines 71-73: is there a reference to cite for this sentence "The validation of this method showed a good agreement on daily and monthly levels; however, various sources of uncertainties have been identified, concerning mainly the use of NN especially during high irradiance atmospheric conditions, the COT, and the structure/density of atmospheric parameters in the LUTs"?**

Reply

*This sentence (and the whole paragraph) has been changed after the major comment 3 of referee 1 (Lines 138-142) and we included the reference Kosmopoulos et al., 2018, as follows:*

"More details about the previous version of the SENSE service can be found in Kosmopoulos et al. (2018). In the same publication the validation of this method showed a good agreement on daily and monthly levels; however, various sources of uncertainties have been identified, concerning mainly the use of the NN especially under high irradiance values, the COT input, and the structure/density of atmospheric parameters in the LUTs."

**Lines 81-83: the meaning of the sentence "However, this first evaluation was based on the satellite-derived COT, so the aim of the current study is to compare the irradiance forecasts with ground-based measurements." is not clear.**

Reply

*The specific sentence has been rephrased (Lines 175-177) and changes have been performed in the whole paragraph 4 of the introduction of the previous version of the manuscript (now paragraph 7 Lines 166-179) in order to make the meaning clearer.*

**Line 96: is there a web link to reach the model and see the GHI estimates? Similarly for NextSENSE2. In case it is useful to add it.**

Reply

The following sentence has been added in Lines 194-195:

"The new version of the SENSE2 system is available as a webservice via https://solar.beyond-eocenter.eu/#solar_short (last access: 2023-12-15)."

**Line 109 and line 112: put a space before "nm".**

Reply

Done.

**Line 129: briefly explain how to correct the surface GHI for sites at higher altitudes than sea level.**

Reply

*The explanation of the correction has been added in Line 226-228:*

"Based on simulations for various atmospheric and surface albedo conditions, Fountoulakis et al. (2021) estimated an average increase of the GHI by 2% per km, which has been also applied to the model output to correct the surface GHI for sites at higher altitudes than sea level."

**Line 145: COT, $R_{eff}$, and LWP a strictly related. The simplest way to see the relation is the formula.**

**LWP=C*r*COT*$R_{eff}$, where C depends on the assumption of the $R_{eff}$ vertical distribution within the cloud, see e.g. Wood and Hartmann, J. Climate, 19, 1748–1764, https://doi.org/10.1175/JCLI3702.1.**

**So if COT is allowed to change in the RTM model simulations with $R_{eff}$ kept fixed, LWP can not remain fixed to 1 g/m$^3$.**

Reply

*We thank the referee for the comment, and we correct this in Lines 273-274, clarifying this relationship by giving the reference of the parameterization used:*

"The COT of the cloud layer is additionally specified at 550 nm, which leads to an adjustment of the liquid water content default value of 1 g/cm$^3$, using the parameterization by Hu and Stamnes (1993)."

**Line 147: the cloud cover fraction is one of the RTM input variable. How is it treated in the simulation of the LUTs?**

Reply

*In our RTM simulation cloud cover fraction equals 100% and only Cloud Optical Thickness varies, as also in previous studies (Taylor et al. 2016, Kosmopoulos et al., 2018). To clarify this aspect of our simulations, a sentence was added in Lines 274-276 of the new version of the manuscript.*

*"*Finally, for the libRadtran simulations homogeneous layer clouds were used, meaning cloud cover fraction value of 100%, which is one of model limitations, since assuming totally cloudy pixels is not always correct for low values of COT (Mueller et al. 2009).*"*

**Line 195: Are there any approaches different from the persistence one to account for modifications in the cloud optical and physical properties?**

Reply

*In Pelland et al. (2013) other common reference forecasts that can be used as benchmark against which to evaluate forecast are provided, which are those based on climate normal and simple autoregressive methods. The reference has been added in the revised version of the manuscript (Line 331) for more details.*

**Line 204: some little information and reference for the two non-BSRN sites of Athens and Thessaloniki may be added.**

Reply

*The information for the two non-BSRN sites of Athens and Thessaloniki has been added in Lines 350-355 of the new version of the manuscript.*

*"*The GHI records that are available at the two Greek stations (1951 – present in Athens, 1993 – present in Thessaloniki) are among the longest continuous high quality GHI records at the Eastern Mediterranean Basin, an area where BSRN data are not available for the period of this study. The pyranometers in Athens and Thessaloniki are calibrated regularly and the GHI measurements have been subjected to quality control before being used in the study. More information for the GHI datasets at the two stations can be found in Bais et al., (2013) for Thessaloniki, and Kazadzis et al, (2018) for Athens.*"*

**Line 208: How is the clear-sky GHI derived for non-BSRN sites? Do authors know how well the Ieichen-Perez clear sky model performs? Did they estimate the deviations compared to GHI measurements in cloud-free conditions?**

Reply

*The clear-sky GHI for the non-BSRN sites was calculated with the same way as for the BSRN stations, by following the methodology described in Yang (2019) and by*

*adjusting the functions of the SolarData v1.1 R package for the non-BSRN stations. A more detailed explanation is provided now in Lines 346-347, as follows:*

"Using the same methodology, the Ineichen-Perez clear sky model values were also computed for the non BSRN station data, by adjusting the functions of the SolarData v1.1 R package for the non-BSRN stations."

*The selection of the Ineichen–Perez clear sky model in the SolarData v1.1 R package is justified by Yang (2018): it is one of the most popular models due to its simplicity (requires only site's altitude and Linke turbidity factor as model inputs) and it was found to be among the best performing models according to the literature. The clear sky model was evaluated by Ineichen (2006) against 16 independent data banks covering 20 years/stations for a large range of altitudes and different climates and he found a MBE of -6W/m² (-1%) which was consistent with our findings when we estimated the deviations for our datasets (a MBE of -9.3W/m² or -1.5%).*

**Equation 6: rRMSE$_{CMV}$ and rRMSE$_{pers.}$ are not introduced.**

Reply

*The relative version of the metrics has been introduced in Lines 376-377 of the new version of the manuscript:*

"The relative values of those metrics rMBE and rRMSE were obtained with respect to the mean of the observed values of GHI."

*and after equation 6 the rRMSE$_{CMV}$ and rRMSE$_{pers}$ have been introduced in Line 382.*

"where rRMSE$_{CMV}$ and rRMSE$_{pers}$ are the relative RMSE of the CMV and persistence forecasting models, respectively."

**Line 242: "due to the limitations in the field of view of the satellite". Explain.**

Reply

*The sentence has been changed in Lines 387-388, in order to better justify the applied threshold related to the highly uncertain satellite cloud retrievals under those conditions:*

", because for higher SZAs the accuracy of the satellite cloud retrievals degrades."

**Line 305: "CMF>0.9" is "CMF≥0.9".**

Reply

It has been changed in the new version of the manuscript.

**Line 308: use "0.4<CMF<0.9" "instead of "CMF <0.9 and >0.4". This is valid for the rest of the manuscript.**

Reply

It has been changed in the new version of the manuscript.

**Line 309: change "the lowest values of measured GHI are found (<250 W/m$^2$)" with "the largest occurrence of small measured GHI values (<250 W/m$^2$) are found".**

Reply

The sentence has been changed according to the suggestion (Lines 490-491):

"In the latest category, the largest occurrence of high deviations at low measured GHI values (<250 W/m$^2$) is found."

**Lines 310-311: again, how do authors support the MSG COT underestimation? If it effect is more evident for high latitude sites, this should be shown.**

Reply

*We agree with the reviewer that we are not talking about "underestimation of satellite COT" since there is not a direct analysis for COT, so this was corrected throughout the manuscript, in line also with the second major comment of the reviewer. We moved the discussion of the reasons of the modeled GHI bias for different conditions in cloudiness later at Section 3.1.3 where the satellite cloud information of CMFmsg is evaluated against ground-based CMF (lines 544 -588).*

**Line 327: report the MBE.**

Reply

", with MBE -28.1 W/m$^2$ or -4.4%" *has been added in Line 520 of the new version of the manuscript.*

**Line 335: until now the authors have not mentioned the 3D effects of clouds and the fact that these cannot be reproduced with 1D models, especially in conditions of partial cloud cover. They should mention this as a limitation and cite the appropriate references.**

Reply

*The respective discussion and the citation of the appropriate references concerning the limitation of using 1D RT models that cannot reproduce 3D effects of clouds have been added in the new version of the manuscript (lines 560-569).*

**Figure 7a: the figure could be larger and the text inside the graph is hard to read.**

Reply

*Old Figure 7 now is Figure 8: it has been updated in order to enlarge Fig.8a and increase the size of the text in the graph.*

**Line 377: the authors mean that the MBE and RMSE are improved after correction, as it is obvious.**

Reply

*The sentence has been changed according to the suggestion of the reviewer (Lines 612-613).*

**Line 383: I would have expected a greater increase in cases with GHI differences within ±50 W/m² after correction.**

Reply

*This could be attributed to two reasons:*

*The first reason could be the fact that the correction was applied only for CMFmsg bins 0.3-0.8, which correspond to the bins that the mean difference in CMF reach its maximum along with low standard deviation for stations used to calculate the correction factor. This "bell-shaped curve" of the CMF bias has also been reported in other studies (e.g. Marie-Joseph et al., 2013) and it was decided only those bins to be included in the correction. However, there are still other sources of bias.*

*The other reason could be that this is a statistic that corresponds to all stations, but the correction wasn't successful for all of them. Specifically, MBE and RMSE have been significantly improved for stations of high cloudiness (e.g. Lerwick) and for all time scales (from 15min to monthly) and to demonstrate this we updated Table 3 (Lines 631-632) by including all the relevant information (in combination also with referee's 1ˢᵗ major comment "**the results of this work should be compared with those of similar studies**"). For Tamanrasset the statistics get worse after the correction, and in combination with all other sources of bias, the increase in cases with GHI differences within ±50 W/m² after correction wasn't so high as expected.*

**I suggest a general review of the English language.**

*The new version of the manuscript has been revised for the English language along with other changes.*

References

[revised manuscript text omitted]